# Mutation in *FBXO32* causes dilated cardiomyopathy through up-regulation of ER-stress mediated apoptosis

Nadya Al-Yacoub[1,2], Dilek Colak [3], Salma Awad Mahmoud[1], Maya Hammonds [4], Kunhi Muhammed[1], Olfat Al-Harazi[3], Abdullah M. Assiri[2,5], Jehad Al-Buraiki[6], Waleed Al-Habeeb[7] & Coralie Poizat[1,4,8 ✉]

Endoplasmic reticulum (ER) stress induction of cell death is implicated in cardiovascular diseases. Sustained activation of ER-stress induces the unfolded protein response (UPR) pathways, which in turn activate three major effector proteins. We previously reported a missense homozygous mutation in *FBXO32* (*MAFbx, Atrogin-1*) causing advanced heart failure by impairing autophagy. In the present study, we performed transcriptional profiling and biochemical assays, which unexpectedly revealed a reduced activation of UPR effectors in patient mutant hearts, while a strong up-regulation of the CHOP transcription factor and of its target genes are observed. Expression of mutant FBXO32 in cells is sufficient to induce CHOP-associated apoptosis, to increase the ATF2 transcription factor and to impair ATF2 ubiquitination. ATF2 protein interacts with FBXO32 in the human heart and its expression is especially high in *FBXO32* mutant hearts. These findings provide a new underlying mechanism for *FBXO32*-mediated cardiomyopathy, implicating abnormal activation of CHOP. These results suggest alternative non-canonical pathways of CHOP activation that could be considered to develop new therapeutic targets for the treatment of *FBXO32*-associated DCM.

[1] Cardiovascular Research Program, King Faisal Specialist Hospital & Research Centre, Riyadh, Saudi Arabia. [2] Comparative Medicine Department, King Faisal Specialist Hospital & Research Centre, Riyadh, Saudi Arabia. [3] Biostatistics, Epidemiology and Scientific Computing Department, King Faisal Specialist Hospital & Research Centre, Riyadh, Saudi Arabia. [4] Masonic Medical Research Institute, Utica, NY, USA. [5] College of Medicine, Al Faisal University, Riyadh, Saudi Arabia. [6] Heart Centre, King Faisal Specialist Hospital & Research Centre, Riyadh, Saudi Arabia. [7] Cardiac Science Department, King Saud University, Riyadh, Saudi Arabia. [8] Biology Department, San Diego State University, San Diego, CA, USA. ✉email: cpoizat@mmri.edu

Heart failure remains a major health burden affecting more than 26 million individuals worldwide with an increasing prevalence[1]. Under physiological stress, the heart increases in size to reduce wall tension and maintain cardiac function[2]. Sustained cardiac stress, however, leads to a pathological growth of the heart progressing to heart failure[3]. Maladaptive cardiac remodeling is mediated by the activation of various signaling pathways that regulate the cardiac proteome through a tight control of protein synthesis and protein degradation[4,5]. Perturbation of the endoplasmic reticulum (ER) function due to accumulation of unfolded proteins, a process termed "ER stress", triggers the unfolded protein response (UPR)[6]. Under physiological conditions, the UPR activates cytoprotective pathways to maintain ER homeostasis[7]. However, oxidative stress, ischemia, or genetic mutations can trigger sustained activation of the ER stress resulting in maladaptive processes such as cardiomyopathies[8–10].

The UPR is regulated by three major effector proteins including protein kinase R-like ER kinase (PERK), activating transcription factor 6 (ATF6), and inositol-requiring enzyme 1 (IRE1), which under normal condition are all maintained in an inactive state through binding to the glucose response protein 78 (GRP78). Upon stress, the UPR response is initiated by the dissociation of GRP78 from the ER transmembrane sensors, which leads to the oligomerization and activation of three downstream UPR effector proteins[11,12]. Oligomerization of PERK facilitates autophosphorylation of PERK cytoplasmic kinase domain, which activates the downstream effector eukaryotic translation initiation factor 2α (eIF2α) through phosphorylation[13]. This in turn attenuates protein synthesis to restore cell homeostasis and elevates the translation of ATF4, a transcription factor that enhances the expression of a wide range of cytoprotective genes[14,15]. ATF6, the second ER sensor, translocates to the Golgi complex following activation where it is cleaved in a 50 kDa fragment (p50) released into the cytosol. p50 subsequently translocates to the nucleus where it regulates the expression of UPR genes involved in protein folding such as the ER chaperone GRP78[16,17]. The third UPR pathway activates the ER sensor inositol requiring kinase 1α (IRE1α), which induces the splicing of mRNA encoding the transcription factor X-box-binding protein 1 (XBP1)[18]. Spliced XBP1s translocates into the nucleus where it binds to the promoter of several genes including UPR chaperones and ER-associated degradation (ERAD) components[19–21].

Under sustained or severe stress, the UPR facilitates the induction of the ER stress-associated cell death[22–24]. The most characterized ER stress-associated apoptosis pathway is mediated by the transcription factor C/EBP homologous protein (CHOP) also known as DNA damage-inducible transcript 3 (DDIT3). CHOP activates apoptosis by regulating the expression of pro- and anti-apoptotic proteins such as BCL-2, BCL-XL, BIM, and GADD34[25]. While CHOP is weakly expressed in the absence of stress[21,26], elevated ER stress in general and dysregulated CHOP protein activity in particular, have been linked to various human diseases including inflammation, metabolic disorders and cardiovascular diseases[27,28]. Prolonged ER stress increases both the IRE1α- and ATF6-UPR pathways, which induce CHOP expression via spliced XBP1 and cleaved ATF6, respectively[20,29]. CHOP expression is also induced through the PERK-eIF2α-ATF4 axis[30,31].

Misfolded proteins are removed by proteolytic systems such as the ubiquitin/proteasome system (UPS), a multistep process in which the substrate specificity is determined by E3 ubiquitin ligases[32]. FBXO32 is a muscle-specific ubiquitin-E3 ligase also known as Atrogin-1/MAFbx, that regulates Akt-dependent cardiac hypertrophy (physiological) and calcineurin A-dependent (pathological) cardiac hypertrophy in mice[33,34]. *Atrogin-1* deficient mice have a cardiac aging phenotype due to dysregulated autophagy caused by aggregation of misfolded proteins[35]. In contrast, transgenic mice with cardiac-specific overexpression of *Atrogin-1* exhibit reduced fibrosis and less collagen deposition in the aged heart[36]. We previously discovered the first homozygous missense mutation in *FBXO32* (c.727G>C, p.Gly243Arg) in humans. The mutation impairs the activity of the Skp1-Cullin-F-Box containing complex (SCF), which as a result dysregulates the autophagy/lysosomal system causing an early-onset dilated cardiomyopathy (DCM)[37]. Whether the *FBXO32* mutation affects the UPR is unknown. Therefore, the objective of the current study was to investigate the role of the UPR in the cardiomyopathy caused by the *FBXO32* mutation. Transcriptional analysis and protein expression analysis in the heart of the patients with the *FBXO32* mutation revealed profound changes in key regulators of the UPR system. While patient hearts expressing the mutant FBXO32 protein unexpectedly showed either an absence or reduced activation of the three UPR branches, they show a strong upregulation of CHOP-mediated apoptosis and abnormally high levels of the ATF2 transcription factor, which binds FBXO32 and whose expression is abnormally high in mutant *FBXO32* hearts and in cells expressing mutant *FBXO32*. These findings provide a new mechanism by which the *FBXO32* mutation causes early-onset cardiomyopathy in human.

## Results

**Major pathways dysregulated in mutant *FBXO32* hearts**. FBXO32 is a muscle-specific ubiquitin ligase belonging to the SCF complex enriched in heart muscle. Gain and loss of function studies in mice have shown a role of FBXO32 in muscle atrophy, pathological cardiac hypertrophy, and cardiomyopathy due to premature cardiac aging[34,35,38–40]. In a large consanguineous family from Saudi Arabia, we discovered that patients who are homozygous for the FBXO32-Gly243Arg mutation develop advanced heart failure. The clinical evaluations of the patients were done by experienced cardiologists and their genetic and clinical relevance has been reported[37]. The family pedigree is shown in Supplementary Fig. 1. Only patients homozygous for the *FBXO32* mutation (patients IV.4, IV.5, IV7, IV.8; Supplementary Fig. 1) developed early-onset dilated cardiomyopathy, due to abnormal accumulation of selective autophagy proteins[37]. The *FBXO32* mutation affects a highly conserved amino acid. In silico analysis using PolyPhen-2, MutationTaster, SIFT, and PHRED predicted that the variant is pathogenic or disease causing[37]. We used three additional prediction algorithms including CADD, GERP, and REVEL, which gave scores of 25.8, 4.95, and 0.658, respectively, and therefore further confirmed that the *FBXO32* variant is disease causing or deleterious. To gain further insights into the mechanisms by which the *FBXO32* mutation causes heart failure, we performed a genome-wide expression profiling in the heart of one patient carrying the *FBXO32* mutation (patient IV.7; Supplementary Fig. 1), which was the only explanted heart available at the time of the analysis. We also assessed gene expression in three control human hearts, four idiopathic dilated hearts (IDC) and in the heart of one patient from an unrelated family carrying a homozygous mutation different than the FBXO32-G243R mutation, causing DCM (Family B). The unsupervised hierarchical clustering in two-dimensions (samples and genes) as well as the principal components analysis (PCA) revealed that the heart samples clustered into four groups with IDC hearts displaying similar gene expression pattern to that from the heart of Family B (Fig. 1a, b; Supplementary Fig. 2). Strikingly, gene expression pattern in the heart of Family A with the *FBXO32* mutation was very distinct from the other groups of hearts (Fig. 1a). Analysis of the data revealed 4806 genes differentially expressed and unique to the

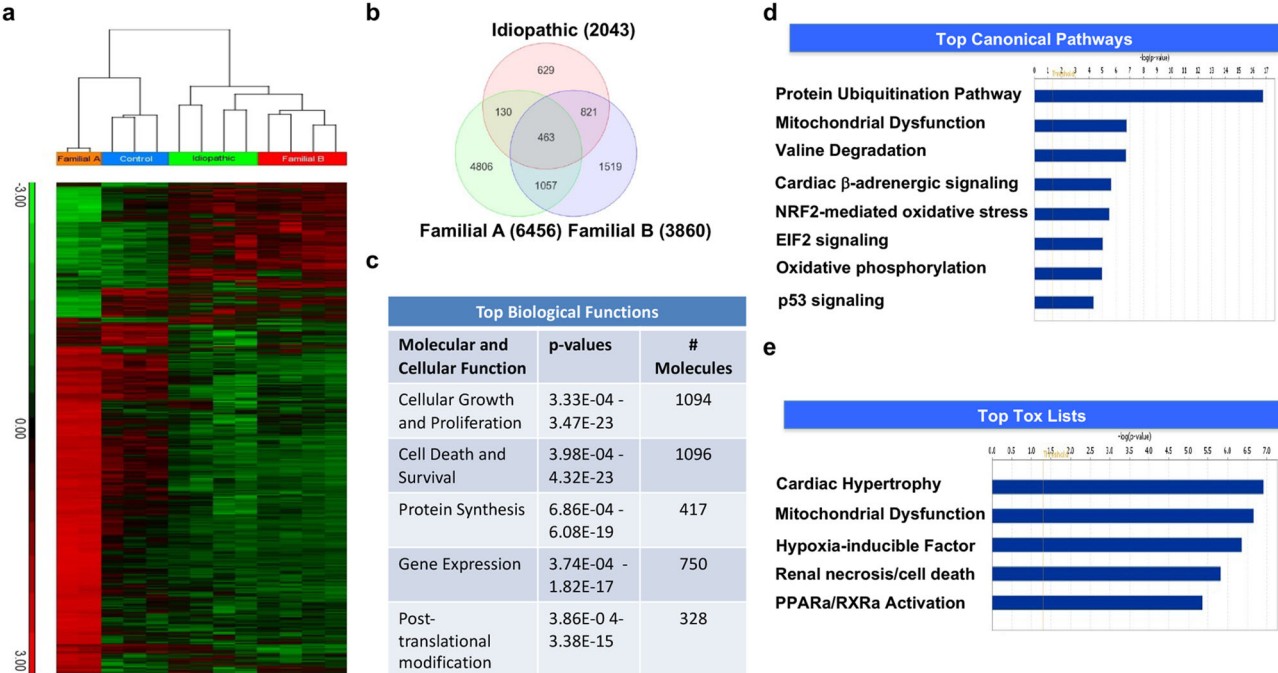

**Fig. 1 Global gene expression analysis of differentially regulated genes in the four groups of hearts. a** Two-dimensional hierarchical clustering of significantly dysregulated genes in heart samples from FDC patients (Familial A) carrying the *FBXO32* mutation, FDC from a second family with a different genetic mutation, and idiopathic heart samples compared to control hearts. Red and green denote highly and weakly expressed genes, respectively. For Family A and B, samples are from one heart run in duplicates and quadruplicates, respectively. Control and idiopathic dilated hearts (IDC) are from 3 and 4 different individuals, respectively. **b** Venn diagram illustrating genes significantly dysregulated in patient group in comparison to the control group. Circles illustrate the number of genes common or unique to each specific group of hearts. Of the transcripts significantly dysregulated, 4806 genes were unique to Family A with the *FBXO32* mutation. **c** Top enriched biological processes, functions, and pathways in the *FBXO32* mutant heart. **d**, **e** IPA functional analysis of the differentially expressed genes in the mutant FBXO32 heart of Family A. The Graphs represent top significant canonical pathways (**d**) and toxicology functions (**e**). *X*-axis indicates the significance (–log10(*p*-value)) of the functional association. The threshold line represents a *p* value of 0.05.

heart with the *FBXO32* mutation (Family A) (Fig. 1b). Among them, 3731 genes were upregulated and 1075 genes were down-regulated (Supplementary Fig. 3). Ingenuity Pathways Analysis was used to identify significantly altered functions and pathways associated with the differentially expressed genes specifically in the *FBXO32* mutant heart. The top biological functions found to be enriched in the *FBXO32* mutant heart included cellular growth and proliferation (1094 genes), cell death and survival (1096 genes), and protein synthesis (417 genes) (Fig. 1c). Top canonical pathways significantly altered in the mutant *FBXO32* heart included mitochondrial dysfunction (Fig. 1d). Consistent with the role of FBXO32 in heart failure, toxicity annotation in IPA showed that cardiac hypertrophy ranked number one in the mutant *FBXO32* heart, followed by mitochondrial dysfunction (Fig. 1e). Interestingly, cell death was among the top five in the biological functions enriched in mutant *FBXO32* hearts. Next, we performed a gene ontology (GO) enrichment analysis using the DAVID functional annotation tool that groups genes according to their ontology. Results showed enrichment of several biological processes including mitochondrial, cell death, and apoptotic process in the mutant *FBXO32* heart (Supplementary Data 1). In addition, the analysis revealed 70 genes with roles in the ER stress response dysregulated in *FBXO32* mutant hearts (Supplementary Data 2). Taken together, these data demonstrated that mito-chondrial function and apoptotic cell death are dominant pro-cesses altered in the heart of the patients carrying the *FBXO32* mutation. Furthermore, the analysis revealed dysregulation of many genes of the ER stress pathway in the mutant *FBXO32* heart.

**Dysregulation of ER-stress proteins and upregulation of CHOP and of its target genes in mutant *FBXO32* hearts.** The ER is a critical organelle which communicates with mitochondria to ensure the maintenance of cellular homeostasis. Because pathways found to be altered in the mutant *FBXO32* heart from our transcriptional analysis included the ER or ER stress response, which is increasingly recognized as playing a role in many cardiomyopathies, we decided to focus our attention on the ER stress pathway and investigate its role in the cardiomyopathy due to the *FBXO32* mutation. Therefore, we next examined the expression of key ER stress proteins including GRP78 and com-ponents of the three ER effector proteins in the three groups of hearts using western blot analysis. For this analysis, we were able to include the heart of another patient from Family A (patient IV.5), who was homozygous for the *FBXO32* mutation. Western blot analysis showed a much lower GRP78 protein level in the two mutant *FBXO32* hearts of Family A (~50% reduction) compared to both controls and IDC hearts (Fig. 2a, b). Similarly, the two *FBXO32* mutant hearts showed reduced activation of the three arms of the UPR as illustrated by a significant decrease of XBP1s within the IRE-1 pathway, reduced expression of phos-phorylated eIF2α within the PERK branch, and reduced cleaved ATF6 (p50), which in fact was only detected in control and IDC hearts (Fig. 2a, b). Next, we assessed the expression of CHOP (C/EBP homologous protein), an ER-stress apoptosis protein known to be induced after prolonged stress. Strikingly, prominent CHOP induction was exclusively observed in *FBXO32* mutant hearts, which also displayed increased expression of the CHOP target protein GADD34 (Fig. 2a, b). These results show hyper-

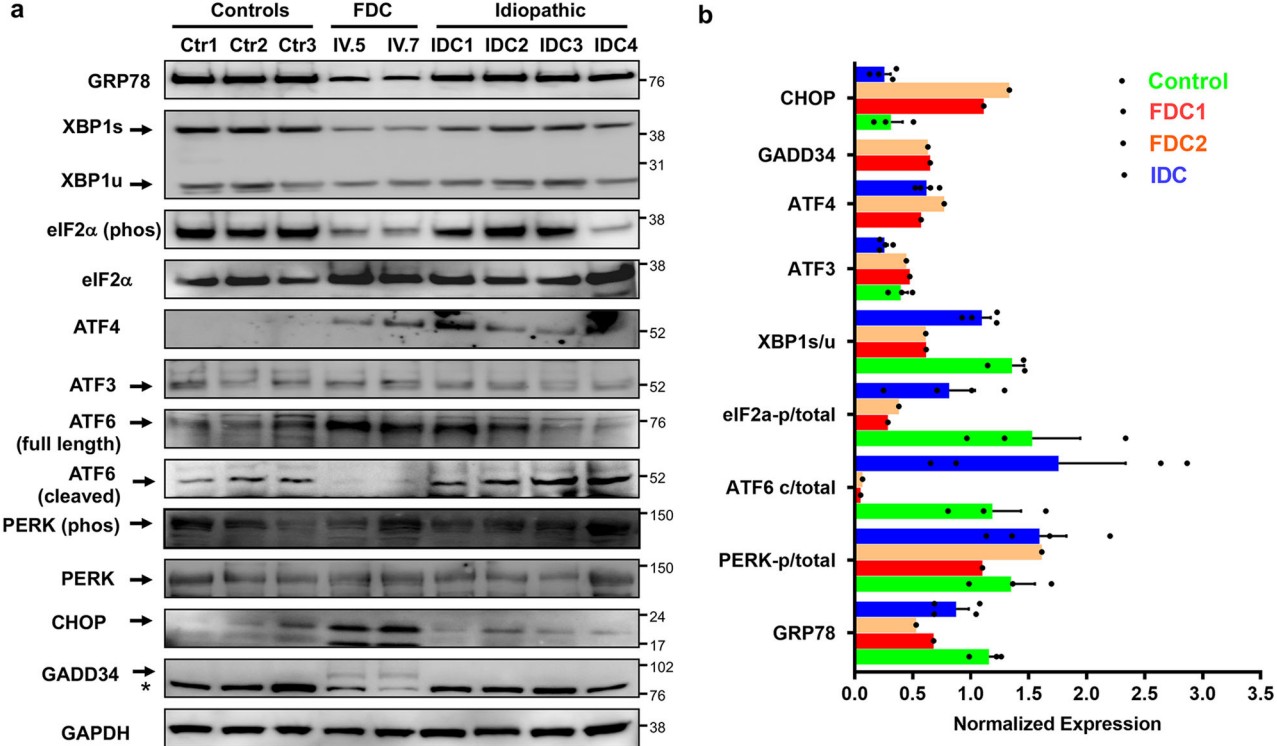

**Fig. 2 Dysregulation of ER-stress proteins and upregulation of CHOP protein in mutant *FBXO32* hearts. a** Immunoblot analysis for major markers of the three UPR canonical signaling pathways performed from control hearts, *FBXO32* mutant hearts (FDC, patients IV.5 and IV.7), and IDC hearts. 25–40 µg of whole protein extract was used to perform the western blot analysis. * indicates non-specific binding of the GADD34 antibody. **b** Quantitative analysis of (**a**); results are normalized against GAPDH and expressed as average ± s.e.m. Control hearts, *n* = 3; FDC (from the two patients carrying the *FBXO32* mutation); IDC: idiopathic dilated hearts, *n* = 4.

activation of ER-stress apoptosis in patient hearts with the *FBXO32* mutation.

Our results showed a prominent induction of CHOP protein in mutant *FBXO32* hearts, which did not occur in control and IDC hearts, suggesting a critical role of CHOP in the cardiomyopathy caused by the *FBXO32* mutation. To validate this, we searched for the expression of CHOP target genes in our microarray data set and compared them from a list of CHOP target genes reported by Han and colleagues[41]. The analysis revealed a total of 111 CHOP target genes that overlapped with published ones, the majority of which were upregulated almost exclusively in *FBXO32* mutant hearts (Supplementary Fig. 4). The list of CHOP target genes from the microarray data analysis with their respective fold changes is provided in Supplementary Data 3. One of the most significantly enriched biological functions associated with the 111 CHOP target genes involved protein synthesis and included three initiation factors (*EIF2S2*, *EIF4G2*, and *EIF5*), several aminoacyl-tRNA synthetases (*IARS*, *EPRS*, *LARS*, *NARS*, *MARS*, and *WARS2*), and 18 genes related to the regulation of gene expression (Supplementary Data 4). IPA network analysis revealed five highly connected focus genes (*BDNF*, *CAV1*, *HSPA5*, *SQSTM1*, and *INSR*) with functions in apoptosis/cell death, insulin signaling, and UPR (Supplementary Fig. 4). Altogether, the induction of *CHOP*, *GADD34*, and a plethora of *CHOP* target genes in mutant *FBXO32* hearts, is indicative of elevated ER-stress and suggest that the ER-stress pathway is a major contributor to the cardiomyopathy caused by the *FBXO32* mutation.

**Upregulation of BIM and activation of caspase-3 in *FBXO32* mutant hearts**. Prolonged ER stress activates CHOP-mediated apoptosis by regulating the expression of BH3-only proteins such

as Bcl-2-like protein 11 (BIM) and B-cell lymphoma-2 (Bcl-2)[25]. Therefore, we assessed the expression of pro- and anti-apoptotic proteins in the three groups of hearts. Consistent with the increased expression of CHOP, pro-apoptotic protein BIM and in particular BimEL protein isoform was high, while anti-apoptotic protein Bcl-2 was low in *FBXO32* mutant hearts compared to control and IDC hearts (Fig. 3a, b). Next, we examined markers of apoptosis activated by BH3-only protein under intracellular stress. Western blot analysis showed increased level of cleaved caspase-3 in the two hearts carrying the *FBXO32* mutation. Furthermore, caspase-3 activation was accompanied by a drastic increase in cleaved Poly (ADP-ribose) polymerase-1 (PARP-1) protein solely in mutant *FBXO32* hearts (Fig. 3a, b). Together, these results show robust activation of the ER-stress-associated apoptosis pathway in the mutant *FBXO32* hearts, suggesting a prominent role of this pathway in the cardiomyopathy caused by the *FBXO32* mutation.

**Increased ATF2 protein expression in mutant *FBXO32* hearts**. Our data show a striking increase of CHOP protein expression in mutant *FBXO32* hearts which is not observed in control and IDC hearts. CHOP can be induced by IRE-1 and ATF6 pathways via spliced XBP1 (XBP1s) and cleaved ATF6 (ATF6c), respectively, and also via the PERK-eIF2α-ATF4 axis. Unexpectedly, our results did not reveal increased XBP1s and ATF6c expression in mutant *FBXO32* hearts compared to controls and IDC hearts. These two proteins were in fact expressed at much lower levels and ATF4 was induced both in *FBXO32* mutant hearts and in IDC. Collectively, these results suggest that CHOP upregulation may be mediated by a non-canonical ER-stress pathway in patients carrying the *FBXO32* mutation. Therefore, we performed gene expression analysis in control hearts, mutant *FBXO32*

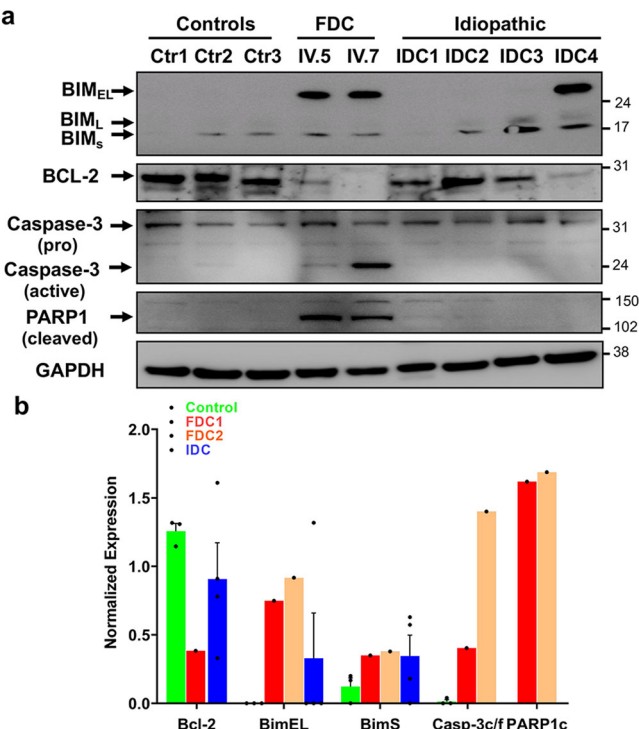

**Fig. 3 Increased CHOP expression plays a role in the activation of apoptosis pathway in FDC hearts. a** The expression of total protein levels of PARP-1 (cleaved), caspase-3 (pro and active), Bim, and BCL-2 were measured by western blot using heart tissue extracts from control hearts, familial hearts from the two patients with the *FBXO32* mutation, and from IDC (idiopathic dilated hearts). Depending on the investigated protein, 25–40 µg of whole protein extract was used to perform the western blot analysis. **b** Quantitative analysis of (**a**); results are normalized against GAPDH and expressed as average ± s.e.m. FDC familial dilated cardiomyopathy, IDC idiopathic dilated cardiomyopathy.

hearts, and IDC hearts, using the human RT² Profiler PCR Array (Qiagen), which includes 84 mRNAs for enzymes playing roles in transcription, protein ubiquitination, and chromatin regulation. Analysis of the data revealed 10 genes significantly upregulated in mutant *FBXO32* hearts compared to IDC hearts ($p \leq 0.05$) (Fig. 4a). Of interest was the observation that the leucin zipper transcription factor ATF2, another member of the cAMP-response-element-binding (CREB) families of transcription factors[42], was expressed at a higher level in the mutant *FBXO32* heart (~4-fold upregulation in *FBXO32* mutant versus IDC hearts). Next, we assessed ATF2 expression at the mRNA and protein level in the three groups of hearts. ATF2 protein level were higher in mutant *FBXO32* hearts compared to controls and IDC hearts (Fig. 4b, c) although no significant difference was observed at the mRNA level between the *FBXO32* mutants and IDC hearts (Supplementary Fig. 5). Based on these observations we hypothesized that the *FBXO32* mutation may regulate ATF2 protein expression. To test this, we analyzed endogenous ATF2 protein expression in HEK293 cells expressing wild-type (WT) FBXO32 or mutant-FBXO32. Immunoblot analysis showed a significantly higher ATF2 protein expression in cells expressing mutant-FBXO32 compared to cells expressing WT-FBXO32 (Fig. 4d, e). Together these results show a significant stabilization of ATF2 protein in the heart of the patients carrying the *FBXO32* mutation and in cells expressing the mutant FBXO32 protein. Altogether with our previous western blot data on other ATF members, our results show a selective increase of ATF2 in the heart of the patients carrying the *FBXO32* mutation.

**FBXO32 interacts with ATF2 in cells and in human hearts, and regulates its ubiquitination**. FBXO32, as a component of the SCF complex, serves as an adaptor that targets specific substrates for ubiquitination. The observation that ATF2 protein is highly expressed in mutant *FBXO32* hearts suggested to us that ATF2 could be a target of the SCF^FBXO32 complex. To test this, we assessed interaction of wild-type and mutant FBXO32 with ATF2 protein in HEK293 cells, which express almost no endogenous FBXO32 protein. WT-FBXO32 or mutant-FBXO32 carrying Flag epitopes were co-transfected with FLAG-tagged ATF2 and FBXO32 was then immunoprecipitated using a specific anti-FBXO32 antibody followed by immunoblotting using an anti-ATF2 specific antibody. IgG antibody was used as control for the immunoprecipitation reaction. As shown in Fig. 5a, ATF2 was efficiently precipitated with WT-FBXO32 and mutant-FBXO32 but not by control IgG. Significant interaction of ATF2 with FBXO32 was also observed in the human heart (Fig. 5b). Using immunofluorescence confocal microscopy, we also detected co-localization of endogenous ATF2 with FBXO32 transduced with a lentivirus vector in primary neonatal rat cardiomyocytes (Supplementary Fig. 6). Together these results show that FBXO32 forms a protein complex with ATF2. Since FBXO32 is part of a ubiquitin ligase complex, we next investigated whether FBXO32 regulates the ubiquitination of ATF2. For this, expression vectors for WT-FBXO32 or mutant-FBXO32 were co-transfected with HA-tagged ubiquitin and FLAG-tagged ATF2 in HEK293 cells, and co-immunoprecipitation was performed using an anti-ATF2 antibody followed by western blot analysis with anti-ubiquitin antibody. Bands of high molecular weight larger than 70 kDa were detected in cells co-expressing WT-FBXO32 and ubiquitin, indicative of ubiquitinated ATF2. In contrast, cells expressing mutant-FBXO32 and ubiquitin showed marginal ATF2 ubiquitination (Fig. 5c). These results indicate that FBXO32 regulates ATF2 ubiquitination which is impaired in cells expressing the mutant-FBXO32 protein.

**Expression of mutant FBXO32 increases CHOP-associated apoptosis**. Our experiments performed in human hearts document enhanced ER-stress-induced apoptosis in mutants *FBXO32* hearts, which suggested to us that FBXO32 may directly regulate ER-stress-induced apoptosis. To test this hypothesis, we transfected WT-FBXO32 or mutant-FBXO32 expression vectors in HEK293 cells treated with the ER-stress inducer dithiothreitol (DTT) and evaluated apoptosis using an ELISA-based assay. Results showed a significantly higher apoptosis in cells expressing mutant-FBXO32 compared to cells expressing WT-FBXO32 (Fig. 6a). Next, we measured CHOP-associated apoptosis. WT- and mutant FBXO32 were expressed at similar levels in cells as shown by the western blot analysis using Flag and FBXO32 antibodies (Fig. 6b). Consistent with our hypothesis, CHOP protein expression was significantly higher in cells expressing mutant-FBXO32 compared to cells expressing WT-FBXO32 after DTT treatment. Furthermore, a significant increase of the CHOP target protein BIM was observed in mutant-FBXO32 cells. Similar to what we observed in *FBXO32* mutant hearts, the BimEL isoform was the one expressed at the highest level in cells expressing mutant-FBXO32. Expression of GADD34 and activation of caspase-3 were also observed in cells expressing mutant-FBXO32 compared to WT-FBXO32 expressing cells although the difference did not reach statistical significance (Fig. 6b, c). Noteworthy, untransfected cells that were used as negative control, exhibited the same expression level for CHOP and BIM as cells over-expressing WT-FBXO32 upon DTT treatment (Fig. 6b, c). This suggests that WT-FBXO32 negatively regulates the expressing of CHOP and subsequently its target gene BIM. Collectively, these

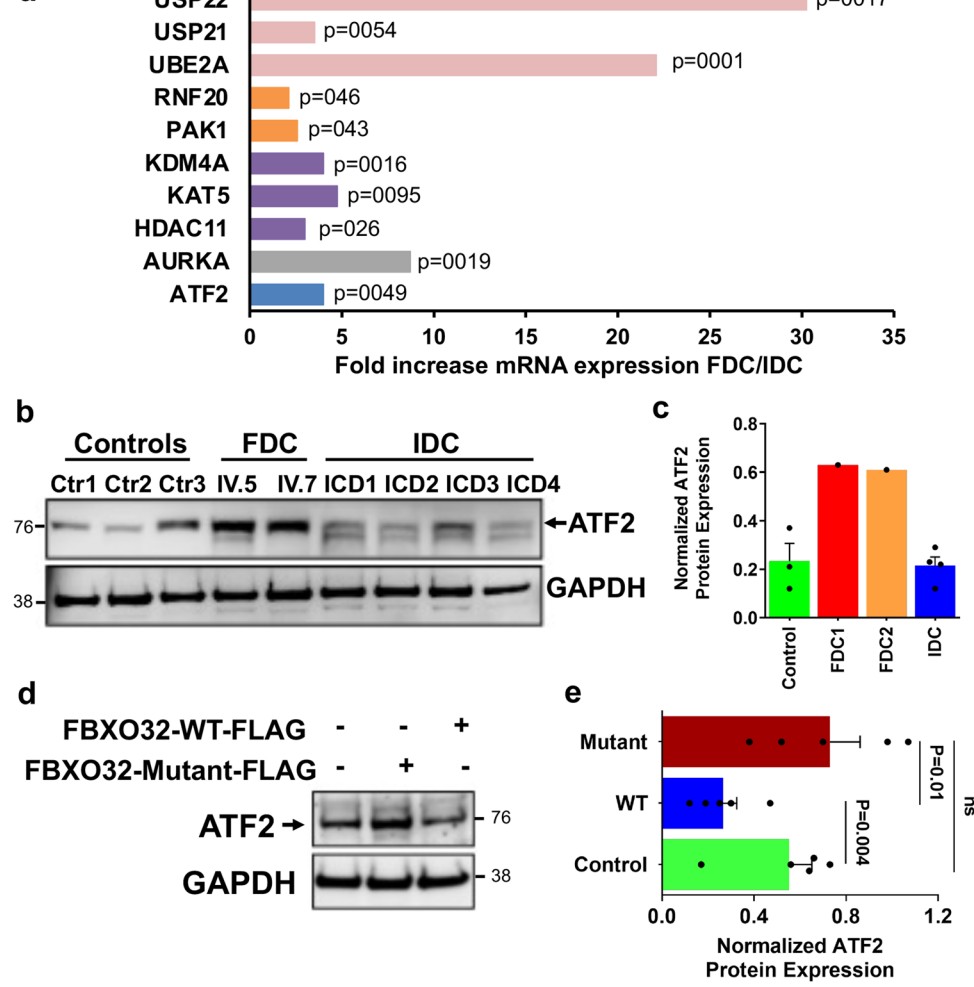

**Fig. 4 Increased ATF2 expression in FDC hearts. a** Total RNA was isolated and reverse transcribed from control hearts ($n = 3$, biological replicates), from the hearts of the patients with the *FBXO32* mutation (Family A, $n = 4$, technical replicates), and from idiopathic dilated hearts ($n = 3$, biological replicates). mRNA expression for enzymes playing roles in transcription, protein stability, and chromatin regulation was measured by qPCR using the RT$^2$ Profiler PCR Array (Qiagen). Transcripts significantly differentially regulated between the *FBXO32* mutant hearts and IDC hearts are shown as fold differences over controls, corrected for 5 housekeeping genes. Significant *p* values calculated using a Student's *t* test are shown. **b** Western blot analysis performed from human heart lysates prepared from control hearts ($n = 3$), the two hearts with the *FBXO32* mutation (FDC), and idiopathic dilated hearts (IDC) ($n = 4$) using a specific anti-ATF2 antibody. GAPDH expression was measured in parallel for loading control. **c** Quantitation of (**b**) using Adobe photoshop CS6-program. Results are expressed as average ± SD and are normalized against GAPDH. **d** Representative immunoblots showing ATF2 protein expression in untransfected HEK293 cells or in cells overexpressing mutant-FBXO32 or WT-FBXO32. **e** Quantitative analysis of (**d**). Results are normalized against GAPDH and expressed as average ± s.e.m., $n = 5$ independent experiments. The statistical significance was determined by Student's *t*-test. Exact *p* values are shown. Ns non-significant.

results corroborate our human findings and show that the *FBXO32* mutation hyperactivates the ER-stress-associated apoptosis response. Altogether, these results suggest that a normal SCF$^{FBXO32}$-complex executes cytoprotective functions by inducing the UPR system and that ATF2 is a previously unknown molecular target of FBXO32.

## Discussion

DCM is a common form of cardiomyopathy with a prevalence now estimated to be 1:250, which is as high as the prevalence of hypertrophic cardiomyopathy[43,44]. DCM is characterized by a dilatation of the cardiac chambers and reduced contractile function of one or both ventricles that can be of genetic cause in about 20–35% of DCM cases. We previously reported that the FBXO32-Gly243Arg mutation impairs the SCF complex which leads to abnormal expression of autophagy proteins[37]. The clinical features of the two patients carrying the homozygous

*FBXO32* mutation used in the present study (patient IV.5 and IV.7) are described in our previous report[37]. Here, we show aberrant activation of the UPR in the heart of two patients with the *FBXO32* mutation, associated with altered expression of mitochondrial genes and upregulation of CHOP indicative of excessive ER-stress apoptosis. This pathway is also activated in cells expressing the mutant FBXO32 protein, underlying the importance of regulated protein homeostasis for the maintenance of heart function.

To understand the mechanisms by which the *FBXO32* mutation causes early-onset heart failure, we compared global gene expression in control, mutant *FBXO32*, and IDC hearts. Consistent with the function of FBXO32 as an E3 ubiquitin ligase and our previous report[37], the *FBXO32* mutation caused a major dysregulation of protein ubiquitination. IPA and DAVID analyses also revealed significant impairment of mitochondrial function and excessive cell death apoptosis in mutant *FBXO32* hearts

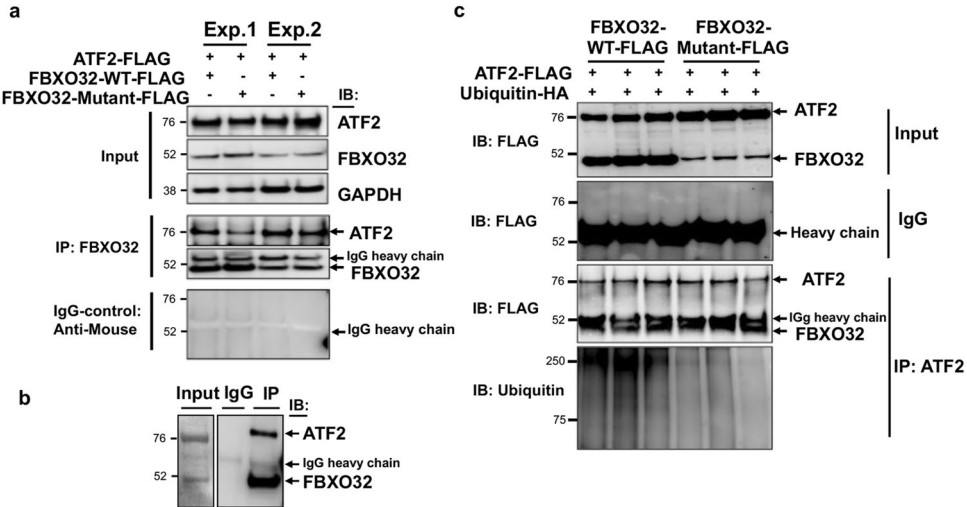

**Fig. 5 FBXO32 interacts and targets ATF2 for ubiquitination. a** FBXO32 interacts with ATF2 in HEK293 cells. Immunoprecipitation performed with FBXO32 antibody or control IgG antibody from HEK293 cells transfected with the indicated plasmids followed by immunoblotting using the indicated antibodies. The results are from two independent experiments. **b** Immunoprecipitation from a human heart showing interaction of ATF2 with FBXO32. IgG-Mouse antibody was used as control. **c** Representative immunoblots showing that FBXO32 regulates ATF2 ubiquitination in HEK293 cells. FLAG-tagged ATF2 was co-transfected with HA-tagged ubiquitin (HA-Ub) and WT-FBXO32 or mutant-FBXO32 in HEK293 cells. Cell lysates were immunoprecipitated with anti-ATF2 antibody and immunoblotted with the indicated antibody. Data are representative of 3 independent transfections for WT-FBXO32 and mutant-FBXO32 ($n = 2$–3 replicates).

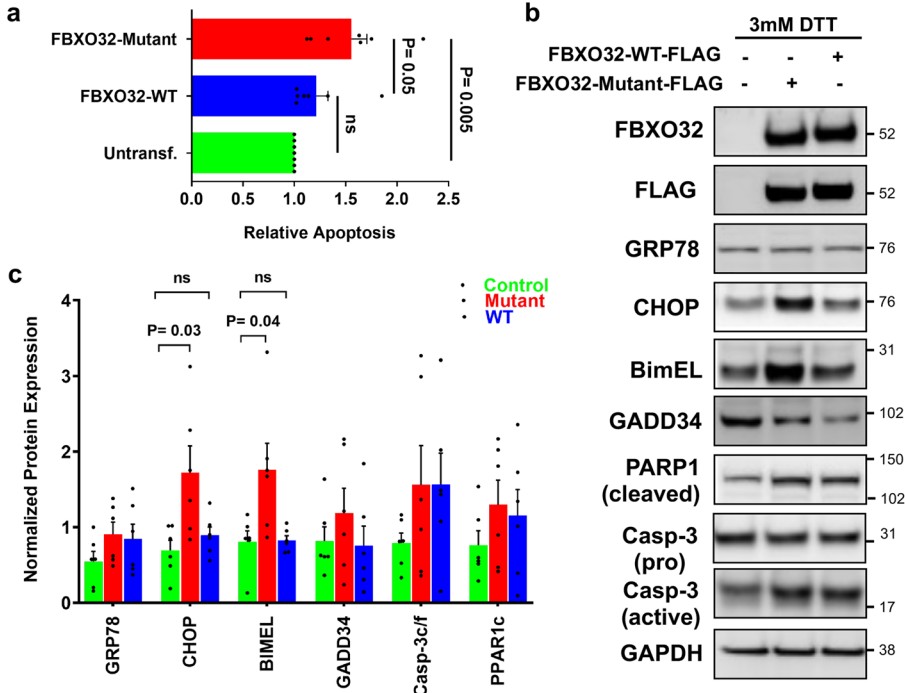

**Fig. 6 Expression of mutant FBXO32 induces apoptosis and increases CHOP-mediated apoptosis in HEK293 cells. a** Increased apoptosis in cells expressing mutant-FBXO32. HEK293 cells were transfected with FBXO32-WT or FBXO32 mutant and 30 h post transfection, DNA fragmentation indicative of apoptosis was measured using an ELISA assay. Results are expressed as relative values to untransfected cells. Data are expressed as average ± s.e.m. from $n = 7$ independent experiments. The statistical significance was determined by Student's $t$-test. $p$ values are shown; $p < 0.05$ was considered significant; ns non-significant. **b** Western blot analysis for CHOP-associated apoptosis pathway using protein extracts from untransfected cells, cells overexpressing mutant-FBXO32 or WT-FBXO32. 24 h post transfection, cells were treated with 3 mM DTT for additional 24 h before being harvested. Western blot analysis was performed using the indicated antibodies. **c** Quantitative analysis of (**b**); results are normalized against GAPDH and expressed as average ± s.e.m. from $n = 5$ independent experiments. $p$ values are shown; $p < 0.05$ was considered significant; ns non-significant.

compared to the other groups of hearts. Genes regulating the ER-stress response were particularly enriched in mutant *FBXO32* hearts. Assessment of key regulators of the ER stress pathway showed a reduced expression of the UPR chaperon GRP78 in the two mutant *FBXO32* hearts that was not observed in controls and IDC hearts. GRP78, a master regulator of the UPR, plays an important role in the heart. Increased GRP78 expression is usually indicative of a prolonged UPR activation and has been reported in DCM and cardiac hypertrophy[45,46]. Unexpectedly, our results showed that *FBXO32* mutant hearts express lower and not more GRP78. Therefore, it is likely that the UPR is activated as an early protective response which then declines under prolonged stress, therefore contributing to heart failure. In support of this, cardiac-specific deletion of GRP78 in adult mice leads to pathological cardiac remodeling associated with increased apoptotic cell death causing early mortality[47]. In contrast, GRP78 overexpression protects against ischemia-reperfusion injury[48]. Therefore, based on the cardioprotective role of GRP78 in vivo, it is reasonable to speculate that the downregulation of GRP78 protein observed in *FBXO32* mutant hearts contributes to the cardiomyopathy rather than represents a secondary compensatory effect.

The active UPR system executes its function through three signaling pathways initiated by the dissociation GRP78 from UPR effector proteins[7]. Examination of key markers of the three UPR branches revealed a striking downregulation of phosphorylated eIF2α and spliced XBP1, and almost no expression of cleaved ATF6 in *FBXO32* mutant hearts. These results suggest either a lack of activation of canonical UPR pathways in *FBXO32* mutant hearts or an accelerated UPR response caused by sustained ER stress. Among these two possibilities, we favor the later because under conditions of sustained or severe ER stress, the UPR system triggers ER-associated apoptosis through transcriptional activation of CHOP[49], which in our study, was highly induced only in *FBXO32* mutant hearts. CHOP-mediated apoptosis is associated with a wide range of diseases including chronic myocardial ischemia and heart failure induced by pressure overload. CHOP-deficient mice subjected to aortic constriction developed less cardiac hypertrophy, fibrosis and showed less apoptotic cell death indicating that CHOP activation is an important mediator of pathological cardiac hypertrophy[45,50]. Our data also show increased expression of many CHOP gene targets and of the CHOP target protein GADD34 in *FBXO32* mutant hearts. Furthermore, expression of mutant FBXO32 in cells was sufficient to sensitize cells to apoptosis and induce CHOP protein. Based on these observations, we conclude that the *FBXO32* mutation is sufficient to induce severe ER stress. This in turn accelerates the cardiomyopathy leading to early-onset heart failure in patients carrying the *FBXO32* mutation.

The UPR and the autophagy/lysosomal systems are two conserved mechanisms that act in concert to degrade unwanted misfolded proteins. After activation, CHOP regulates several pro- and anti-apoptotic proteins including GADD34, BIM, and BCL-2 implicated in the regulation of cell death by apoptosis[28]. Our results suggest that the cardiomyopathy induced by the *FBXO32* mutation results from the impairment of several pathways converging to ER stress and apoptosis. In support of this, the expression of pro-apoptotic BIM was especially high in the mutant *FBXO32* hearts. Among the three major BIM isoforms, BimEL showed the strongest increase in mutant *FBXO32* hearts and in cells expressing mutant FBXO32. It has been shown that BIM inhibits autophagy independent of its pro-apoptotic function by interacting with Beclin-1 in a complex with LC8. Among the three BIM isoforms, only BimEL and BimL strongly interact with Beclin-1[51]. In addition, BimEL is an important factor regulating the switch from autophagy to apoptosis during prolonged

starvation[52]. Based on these reports, it is likely that the upregulation of BimEL in *FBXO32* mutant hearts provides a link between the activation of apoptosis and the impairment of autophagy that occurs in the mutant *FBXO32* hearts[37].

A substantial decrease of anti-apoptotic BCL-2 protein was observed in mutant *FBXO32* hearts. Members of the BCL-2 family control cell death primarily by regulating the intrinsic apoptotic pathway. A disturbance in the balance between pro- and anti-apoptotic BCL-2 members affects the mitochondrial outer membrane permeabilization leading to the activation of caspases and apoptosis[53]. Likewise, mutant *FBXO32* hearts displayed activation of the major effector caspase-3 primarily responsible for the execution of cell death. Active caspase-3 was more robustly increased in patient IV.7 compared to patient IV.5, likely due to a biological difference. Cleaved PARP-1, which is mediated by activated caspase-3, was only detected in *FBXO32* mutant hearts. Furthermore, GADD34, another pro-apoptotic target of CHOP known to promote the dephosphorylation of eIF2α to enhance ER stress-induced apoptosis[28], was solely detected in mutant *FBXO32* hearts. These results suggest that GADD34 may potentiate the intrinsic-apoptosis pathway in mutant *FBXO32* hearts. We searched for genes of the extrinsic apoptosis pathway in our transcriptome data. We could not find a significant dysregulation of TNFα, TNFR1, DR3/4/5, FAS, FADD, TRADD, RIP, or DED, the most important members of this pathway. Only TNFRSF21 (DR6) was differentially regulated in the mutant *FBXO32* heart. Although we cannot exclude the possibility that differences may occur at the protein level, this suggests that the extrinsic apoptosis pathway plays a minor role in *FBXO32*-mediated cardiomyopathy. Together, our findings provide evidence for a strong activation of the intrinsic-apoptosis pathway in patients carrying the *FBXO32* mutation. Apoptosis is a well-known contributor to heart failure[54,55] and collectively, our results implicate that the *FBXO32* mutation activates at least two CHOP-mediated apoptosis pathways leading to premature cardiomyopathy and heart failure.

CHOP is transcriptionally induced during ER stress predominantly through the PERK/eIF2α/ATF4 pathway. The absence of significant induction of ATF4 and ATF6 in *FBXO32* mutant hearts suggests the existence of alternative pathways activating ER-stress-associated apoptosis. Examples of non-canonical signaling pathways regulating ER-stress apoptosis, cardiac hypertrophy, and heart failure have been reported[56]. Several members of the CREB/ATF transcription factor family interact with the CHOP promoter to transcriptionally regulate its expression. ATF5 activates while ATF3 represses CHOP expression during arsenite-induced stress[57,58]. ATF3 can also cooperate with ATF4 to activate CHOP expression under various stress stimuli[59,60]. Furthermore, ATF4 and ATF2 activation are required to induce CHOP expression in response to amino acid starvation[61]. Thus, we explored the possibility that ATF2 may be implicated in the ER-stress induced by the *FBXO32* mutation. Our transcriptional analysis combined with our biochemical assays documenting binding of FBXO32 to ATF2 in the human heart identified ATF2 as a new cardiac substrate of FBXO32. Indeed, ATF2 protein accumulated solely in the hearts of the patients with the *FBXO32* mutation. Expression of mutant *FBXO32* was sufficient to stabilize ATF2 protein, which also correlated with increased ER stress and apoptosis. Consistent with the function of FBXO32 as an E3 ubiquitin ligase, our data revealed that FBXO32 regulates ATF2 ubiquitination. Collectively, our data suggest that upregulation of ATF2 due to the *FBXO32* mutation, is associated with activation of the ER-stress apoptosis pathway. ATF2 regulates diverse cellular functions through dimerization with various AP1 family members depending on the cell type and stimuli[62]. Depending on the

stimulus and on its subcellular localization, ATF2 can act as a transcriptional activator or repressor[42,63]. To date, very few studies have explored the role of ATF2 in the pathogenesis of cardiovascular disorders. Desmin-positive protein aggregates correlate with high levels of ATF2, voltage-dependent anion-selective channel protein 1 (VDAC1), and BCL2 Associated X (BAX) in muscle fibers of desminopathy patients[64]. Ischemia/ reperfusion activates the PERK-UPR signaling pathway which correlates with ATF2 upregulation[65]. Increased ATF2 phosphorylation has been reported in response to angiotensin-II and IL-1β implicating ATF2 in vascular endothelial dysfunction[66,67]. Finally, TGFβ induces cardiac hypertrophy by PKC-dependent ATF2 activation[68]. Thus, while our results point toward ATF2 as a novel FBXO32 target, further experiments are required to establish the direct link between ATF2 and the cardiomyopathy caused by the *FBXO32* mutation.

One limitation of our study is with the availability of only two human hearts with the *FBXO32* mutation. Because sample heterogeneity is often an issue with human specimens, we minimized the collecting time of the human hearts by freezing the hearts in liquid nitrogen immediately after surgical removal from the patients, to preserve tissue integrity. Using this strict protocol, variability between explanted hearts was reduced to a minimum. In addition, the effect of the mutation was studied in HEK293 cells, which validated the results observed in the mutant human hearts. The heterozygous state of the mutation was not tested in the current study because our genetic analysis showed that only the patients with the homozygous mutation develop DCM. It would be of interest to assess the effect of the heterozygous mutation in cells. This is beyond the scope of this work and will be the topic of a future investigation.

**Conclusion.** The present study links for the first time a mutation in *FBXO32* with excessive ER-stress-induced apoptosis and DCM. Our findings highlight a critical mechanism underlying the pathogenesis of DCM and suggest that targeting CHOP or perhaps ATF2 should be considered as an approach for the treatment of DCM and in particular the cardiomyopathy due to the *FBXO32* mutation (Fig. 7). Additional research will be necessary to elucidate the precise molecular mechanism by which CHOP inhibits autophagy in dilated cardiomyopathy cases that are linked to ATF2-dependent induction of CHOP-induced apoptosis. Therapies aimed to restore the autophagic flux at an early stage might represent another approach to maintain adaptive autophagy and prevent the maladaptive apoptosis response leading to heart failure.

## Methods

**Human hearts collection.** Control donor hearts used in this study were from patients with no sign of cardiac disease whose hearts could not be transplanted due to ABO mismatch. Four hearts from patients with IDC were used in the study. The two hearts with the *FBXO32* mutation were obtained from two affected individuals (patient IV.5 and IV.7) from a family in Saudi Arabia that was diagnosed with familial dilated cardiomyopathy. Recruitment of the family members, the clinical features of the affected patients, and assessment of their cardiac function are described in detail in our original report[37]. The hearts were collected at the time of heart transplant and immediately frozen in liquid $N_2$ to preserve tissue integrity. For biochemical analysis, a sample of the left ventricle from the same anatomical region was dissected. All participants provided written informed consent under protocols approved by the Institutional Review Board at King Faisal Specialist Hospital & Research Centre (RAC#2100 024 and #2010 020).

**Antibodies and plasmids.** Antibodies for FBXO32 (sc-166806, 1:300 dilution), cleaved PARP-1 (sc-56196, 1:400 dilution), BIM (sc-374358, 1:400 dilution), ATF3 (sc-188, 1:200 dilution), GAPDH (sc-25778 and sc-47724, 1:1000 dilution), GRP78 (sc-376768, 1:1000 dilution), PERK (sc-377400, 1:300 dilution), and ATF2 (sc-187 and sc-242, 1:300 dilution) were purchased from Santa Cruz Biotechnology. Antibodies for full length and cleaved ATF6 (70B1413.1, 1:500 dilution), caspase-3 (31A1067, 1:500 dilution), and XBP1 (NBP1-77681, 1:300 dilution) were from

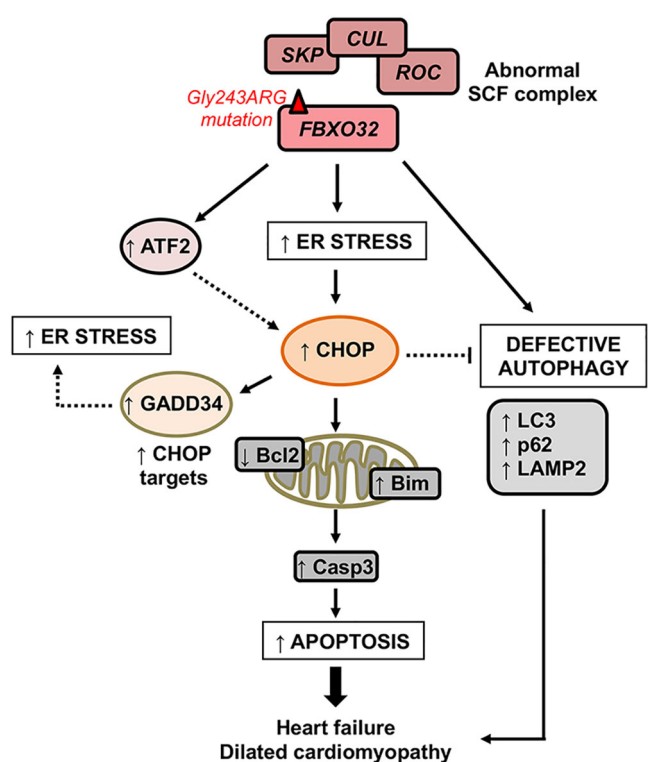

**Fig. 7 Model depicting the role of FBXO32 in the crosstalk between CHOP-dependent apoptosis and autophagy.** Mutation in the FBXO32 E3 ligase activates the ER stress response, which in turn induces the expression of CHOP protein. Increased CHOP expression activates the intrinsic-apoptosis pathway and induces GADD34 expression, which could amplify the apoptotic effect by promoting protein synthesis that leads to increased ER stress. The *FBXO32* mutation increases ATF2 possibly increasing cell death. Together these molecular mechanisms amplify the maladaptive ER stress response and contribute to the development of heart failure. ER endoplasmic reticulum, DCM dilated cardiomyopathy.

Novus Biologicals, LLC. Antibodies for GADD34 (10449-1-AP, 1:200 dilution) and CHOP (15204-1-AP, 1:200 dilution) were from Proteintech. Antibodies for ATF4 (#11815, 1:500 dilution), eIF2α (#9722, 1:500 dilution), p-eIF2α (#9721, 1:500 dilution), Ubiquitin (#3933. 1:500 dilution), and Bcl-2 (#15071, 1:500 dilution) were from cell signaling. Antibody for PERK-Thr980 (bs-3330R, 1:100 dilution) was from Bioss Antibodies Inc. FLAG-Probe (sc-166355; 1:500 dilution) was purchased from Santa Cruz. Mouse (G3A1) mAB IgG1 Isotype control was from Cell Signaling (#5415; 5 μg). For immunofluorescence confocal microscopy, secondary antibodies donkey anti-mouse Alexa Fluor™ 488 (#A21202) and donkey anti-rabbit Alexa Fluor™ 555 (#A31572) were purchased from Thermo Fisher Scientific.

Eukaryotic expression vectors for wild-type FBXO32 carrying a Flag epitope (FBXO32-FLAG) and ATF2 also carrying a Flag epitope (ATF2-FLAG) were purchased from Origene. The mutant FBXO32 flag tagged plasmid was generated using site-directed mutagenesis as described previously[37]. HA-ubiquitin plasmid (#18712) was from Addgene. Third-generation lentivirus vectors were as follows: Plasmid pMD2.g (#12259), pRSV-Rev (#12253), and pMDLg/pRRE (#12251) were purchased from Addgene. The lentivector for wild-type FBXO32 (#RC223661L1) was from Origene and was verified using restriction digest and Sanger sequencing.

**Array hybridization and microarray analysis.** Total RNA was extracted from heart tissues using Trizol reagent (Thermo Fisher Scientific). Affymetrix's Gene-Chip® Human Genome 430 2.0 Arrays were used. Sample handling, cDNA synthesis, cRNA labeling and synthesis, hybridization, and washing were performed according to the manufacturer's instructions. Data normalization was performed using GC Robust Multi-array Average (GC-RMA) algorithm[69]. To determine significant differences in gene expression levels among different groups, ANOVA was performed. Significantly modulated genes between patient and control samples were defined as those with absolute fold change (FC) > 1.5 and *p*-value <0.05. Functional, pathway, gene ontology (GO), and network analyses were performed using Database for Annotation, Visualization and Integrated Discovery (DAVID) Bioinformatics Resources[70] and Ingenuity Pathways Analysis (IPA)

(QIAGEN Inc., https://www.qiagenbioinformatics.com/products/ingenuity-pathway-analysis). A right-tailed Fisher's exact test was used to calculate a p-value determining the probability that the biological function (or pathway) assigned to that data set is explained by chance alone. Statistical analyses were performed by using SAS 9.4 (SAS Institute, Cary, NC) and PARTEK Genomics Suite (Partek Inc., St. Lois, MO, USA). All statistical tests were two-sided and p-value <0.05 was considered statistically significant.

**RNA extraction and RT2 Profiler™ PCR Array.** The Human Epigenetic Chromatin Modification Enzymes RT² Profiler PCR Array profile was used to screen the expression of 84 key genes encoding enzymes known or predicted to modify genomic DNA and histones to regulate chromatin accessibility and therefore gene expression (#330231/PAXX-085Y). Total RNA was isolated from one FDC patient heart, 3 control, and 3 idiopathic hearts using Qiagen RNeasy Fibrous Tissue Mini Kit (74704) following the manufacturer's instruction. One microgram of total RNA was used for reverse transcription using iScript™ cDNA Synthesis Kit (#1708890). The so obtained cDNA was then utilized for the PCR array according to manufacturer's recommendations. All of the real-time PCR reactions were performed with RT2 SYBR Green PCR master mix on CFX96 real-time system (Bio-Rad). Data analysis was conducted using a spreadsheet-based tool downloaded from the QIAGEN website.

**Cell culture, DNA transfection, and treatment.** HEK 293T/17 cells, derivative of the 293T cell line, were purchased from ATCC (CRL-11268). Cells were cultured in DMEM (high glucose) supplemented with 10% fetal bovine serum and 1% penicillin-streptomycin. Expression vectors for wild-type FBXO32, mutant-FBXO32, ATF2 carrying a Flag epitope, and Ubiquitin with an HA tag were co-transfected in HEK293 cells using polyethyleneimine (PEI) transfection method according to the protocol described in ref. [71]. The total DNA amount used was 16 μg for 10-cm dish and 3 μg for one well of a 6-well plate. Plasmids were used at a ratio of 2:1 for FBXO32 and ATF2, and 4:1:1 in case of three plasmid transfection. To induce ER stress 24 h post transfection cells were treated with 3 mM Dithiothreitol (DTT) for additional 24 h and then harvested for protein extraction.

**Measurement of apoptosis.** To quantify apoptosis, cells were plated in a 6-well plate and transfected with WT-FBXO32 or mutant-FBXO32. Thirty hours post transfection cells were harvested and apoptosis was measured using the Cell Death Detection ELISA kit (Roche Diagnostics; # 11920685001) following the manufacturer's instruction.

**Ubiquitination assay.** To measure ATF2 ubiquitination, cells were co-transfected with the indicated plasmids as described above. Thirty hours post transfection cells were harvested and whole-cell extracts were prepared and used for co-immunoprecipitations as described in ref. [72].

**Isolation of primary neonatal rat cardiomyocytes.** Primary neonatal Rat Cardiomyocytes were isolated using the Neomyt kit from Cellutron (Cat# nc-6031). Briefly, 1–2-day-old pups were euthanized by decapitation and the chest was opened to quickly remove the hearts. After a succession of enzymatic digestions, neonatal rat ventricular myocytes (NRVM) were dissociated and collected by pre-plating the fibroblasts for 45 min. Ventricular myocytes were plated in 10-cm dishes coated with SureCoat at a density of $9 \times 10^6$ cells/dish in DMEM containing high glucose, 10% fetal bovine serum (FBS), and antibiotics (100 U/ml penicillin, 10 mg/ml streptomycin solution). The following day, the cell media was removed, and cells were fed with DMEM supplemented with 5% FBS and antibiotics and maintained in a $CO_2$ (5%) incubator at 37 °C.

**Production of recombinant lentiviruses and immunofluorescence microscopy.** Recombinant lentiviruses were produced by transfecting HEK293T cells with pMD2.g (Addgene #12259), pRSV-Rev (Addgene #12253), pMDLg/pRRE (Addgene #12251), and lentivector for wild-type FBXO32 (Origene #RC223661L1) using Lipofectamine 3000 (Invitrogen). A control lentivirus expressing GFP was used as control. Viral titers were determined by measuring the % of GFP-positive cells after FACS analysis or by fluorescence microscopy from serial dilutions of the virus. NRVM plated in 12 well plates on cover slides were transduced with lentiviruses at an MOI of 2.5–5. Forty-eight hours after transduction, cells were permeabilized and immunofluorescence was performed using FBXO32 and ATF2 antibodies diluted in a PBS cocktail (PBS, 1% BSA, 1% Triton X-100) overnight at 4 °C. For Secondary antibody reactions, donkey anti-mouse Alexa Fluor™ 488 or donkey anti-rabbit Alexa Fluor™ 555 (Thermo Fisher Scientific) were used. Coverslips were mounted with Vectashield DAPI containing media (Vector Laboratories) and signals were visualized using confocal microscopy (Zeiss LSM 700).

**Protein extraction and western blot analysis.** Protein extracts from human heart tissue were prepared by grinding the tissue in liquid nitrogen. The powder was then homogenized using sonication in a high salt lysis buffer supplemented with protease inhibitor cocktail (20 mM Hepes pH 7.5, 0.65 M NaCl, 1 mM EDTA, 0.34 M sucrose). Lysates were centrifuged at 12,000 rpm for 15 min and supernatants were recovered. Protein concentration was measured using the Quick Start Bradford 1x Dye Reagent (#5000205 Bio-Rad). Extracts from transfected HEK293T cells were prepared following the same protocol excluding the grinding in liquid nitrogen step.

For immunoblotting, equal amounts of protein (20–50 μg) were separated by sodium dodecyl sulfate-polyacrylamide gel electrophoresis (SDS-PAGE) and transferred to nitrocellulose membrane (GE Healthcare, Amersham) using the semi-dry transfer blotting method. Before blocking if needed membranes were cut into horizontal strips according to the molecular weights of the investigated protein to enable the analysis of different proteins on the same membrane. After blocking with 5% non-fat milk in TBST (20 mM Tris, 150 mM NaCl, pH 7.4, 0.1% Tween-20), membranes were incubated with the indicated primary and corresponding secondary antibodies. Blots were developed using the chemiluminescence ECL detection kit from Pierce (#32132). Images from western blot were captured using a LAS 4000 analyzer (GE).

**Co-immunoprecipitation.** For co-immunoprecipitation experiments, 2 mg of total heart protein lysates were incubated with antibody against FBXO32 or control IgG overnight at 4 °C under gentle rotation in a total volume of 1 ml. The following day, 100 μl of protein A/G agarose was added to the lysate and incubated under rotation for 8 h at 4 °C. Agarose beads were washed three times in washing buffer (0.025 M Tris, 0.15 M NaCl, 0.001 M EDTA, 1% NP-40, 5% glycerol; pH 7.4) and after elution with 2× Laemmli sample buffer (#161-0747, Bio-Rad), immunoprecipitates were analyzed by western blot using the indicated antibodies as described before.

**RT-qPCR.** RNA extraction and cDNA synthesis were performed as described before. Ten nanograms of cDNA were then analyzed by qPCR utilizing the SYBR Green Master mix from (A25742) and the 7500 Fast Real-Time PCR System from Applied Biosystems. Specific primers for GAPDH (F: ACATCGCTCAGACACC ATG; R: TGTAGTTGAGGTCAATGAAG) and ATF2 (F: CTGTTTACCCCCA TTGTCTTG; R: CACACACACTCTCTCTCATTC) were designed.

**Statistics.** Data are presented as average ± s.e.m. Statistical comparisons were performed using unpaired one- or two-tailed Student's t test; $p < 0.05$ were considered statistically significant. Exact p values are shown when appropriate. Graphs were generated using GraphPad Prism software.

**Ethics approval and consents.** The study was carried out in accordance with the principles of the Declaration of Helsinki. Human subjects provided written informed consent under protocols approved by the Institutional Review Board at King Faisal Specialist Hospital & Research Centre (RAC#2100 024 and #2010 020).

## Data availability

Microarray data can be accessed at the following link: https://www.dropbox.com/s/k6609c3i43roiyq/MicroArray%20Data_DCM.xlsx?dl=0. Alternatively, the datasets generated during and/or analyzed during the current study are available from the corresponding author on reasonable request.

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

## Acknowledgements
We thank Ms. Israa Zahid for collecting the human hearts from the Heart Centre at King Faisal Specialist Hospital & Research Centre (KFSHRC), and Ms. Kamar Al-Haffar for performing RNA extraction from the human hearts. We also are thankful to Ahmed Abusaleem, Khalid Al-Khatib, and other members of the Heart Transplant Team at KFSHRC for coordinating the transfer of the explanted hearts. We thank Dr. Gulhan Ercan-Sencicek for performing the in silico prediction of pathogenicity using CADD, GERP, and REVEL tools. This work was supported by institutional funds from King Faisal Specialist Hospital & Research Centre and from the Masonic Medical Research Institute.

## Author contributions
N.A.-Y. designed, performed all biochemical experiments, collected data, and wrote the manuscript. D.C. supervised and performed bioinformatic analysis of microarray data. S. M.A. performed the Qiagen RT2 Profiler™ PCR Array experiment. M.H. produced recombinant lentiviruses and isolated NRVM for immunofluorescence microscopy. K.M. collected the human hearts and prepared the heart protein extracts. A.-H.O. performed data analysis. W.A.-H. and J.A.-B. recruited the family and provided patients clinical data including the cardiology assessment of DCM. A.A. supervised part of the project. C.P. oversaw the project, collected data, and wrote the manuscript. All authors read and approved the final manuscript.

## Competing interests
The authors declare no competing interests.
