## [Peer Review File · Communications Biology]

Reviewers' comments:

Reviewer #1 (Remarks to the Author):

The manuscript by Al-Yacoub et al investigates the consequences of FBXO32 mutation on dilated cardiomyopathy. The authors perform RNA-seq on a single patient sample, and determine that ER-stress-related genes are altered when compared to controls, or other causes of dilated cardiomyopathy. Subsequent investigations on the same sample and on a second sample from the same family confirm, by PCR and Western blot, that markers of ER stress are up-regulated in these patients, which results in activated caspase 3 and apoptosis. The data are interesting and seems to point to a role for this mutation to induce an ER stress response that is mediated by interaction with ATF2. The biggest issue with the study is the reliance on one patient sample, even if confirmation is done in samples from two individuals. At least the cell experiments should have been repeated minimally 4 times, to determine if the observed effects are truly due to the mutation.

1 – In the text the authors mention that GADD34 is increased in the hearts of the mutant patients, but bands in Figure 2 are not well defined. An upper band is only present in the mutant hearts, but a lower band is dramatically decreased. There is no explanation for the lower band or how it relates to the mutation.

2 – The significance of the isoforms of BIM should be discussed.

3 – The small sample size needs to be added to a limitations section.

4 – Is there a reason why protein levels in Figure 2 were not compared to Family B?

5 – Please name the excel supplementary files as described in the text.

6 – Bar graphs are not appropriate, especially since this is a N=2. Dot blots should be shown.

7 – I assume that the two samples from family A in Figure 1 are technical replicates from the same heart. This needs to be disclosed in the figure legends.

8 – The small sample size decreases rigor of the work. Although this is a rare mutation, the authors do provide cell data on the effect of the mutation in cells. It is absolutely necessary that the authors increase the number of cell experiments so that rigor can be increased through proper statistical analysis. As is, this is a subtle suggestion of the role of this mutation on ER stress and apoptosis.

Minor revisions

Line 90-change sever—severe

Line 840-change heats—hearts

In Figure 4C change label from protein expression to ATF2 expression

Reviewer #2 (Remarks to the Author):

In the manuscript entitled "Mutation in FBXO32 causes dilated cardiomyopathy 1 through upregulation of ER-stress mediated Apoptosis", Al-Yacoub et al found a new mutant of Fbxo32 (Gly243Arg) and this single amino acid mutant led to aberrant activation of CHOP and the CHOP associated apoptosis. They also found that ATF2 expression level was elevated in the Fbxo32 mutant heart. They also confirmed the results in 293 cells by overexpressing the mutant Fbxo32. They found an interesting Fbxo32 mutant and its potential functions in heart. I have the following concerns for the authors to be addressed.

1. There are many key experiments performed in 293 cells. The advantage of using 293 cells has been stated by the authors clearly that no endogenous Fbxo32 to compete with the mutant Fbxo32.

However, there may be potential problems by solely doing experiments in 293 cells. The context of 293 cells is drastically different from that of cardiac cells. The key results such as the interaction between Fbxo32 and ATF2, the induction of ATF2 after overexpression of mutant Fbxo32 should be confirmed in cardiac cells.

2. The ubiquitination status of ATF2 after Fbxo32 mutant or WT Fbxo32 overexpression should be checked by ubiquitination Western blot.

3. To determine the occurrence of apoptosis only by Western blot of cleaved caspase 3 is kind of

weak. Other assays such as Annexin-V staining, TUNEL assays should be performed to confirm the occurrence of apoptosis in Fbxo32 mutant cardiac cells.

4. Similarly, Other assays such as Annexin-V staining, TUNEL assays should be performed to confirm the occurrence of apoptosis in 293 cells overexpressing mutant or WT Fbxo32.

5. The authors described in the text that "untransfected cells that were used as negative control, exhibited the same expression level for CHOP and BIM as cells overexpressing WT-FBXO32 upon DTT treatment." The corresponding figure has not been cited.

6. The expression level of GADD34 and Caspase 3 seem to be unstable in untransfected 293 cells. What is the explanation? Is it simply experimental variations? Why is the variation specifically big with these two proteins?

7. Related to question 6, in Fig. 5a expt 1, GADD34 protein level decreased when compared to the untransfected 293 cells; while GADD34 protein level increased when compared to the untransfected 293 cells in expt2. Please explain.

Reviewer #3 (Remarks to the Author):

Associate Editor
Communications Biology
One New York Plaza, Suite 4600
New York, NY 10004-1562
orcid.org/0000-0003-1092-3821
faten.taki@us.nature.com

Dear Associate Editor,

The manuscript entitled: "Mutation in FBXO32 Causes Dilated Cardiomyopathy Through Upregulation of ER-Stress Mediated Apoptosis" by Al-Yacoub and colleagues, describes their original research focused on investigation the mechanism by which the FBXO32 mutation (identified previously) causes advanced cardiomyopathy. They performed transcriptional profiling and biochemical assays in the heart of patients with the FBXO32 mutation and compared with heart samples obtained from patients with idiopathic dilated cardiomyopathy and control donors. They found reduced activation of the three UPR effectors while a strong up-regulation of the transcription factor CHOP and of its target genes. They also claim that they found severe apoptosis specifically in FBXO32 mutant hearts. Then, they expressed mutant FBXO32 in cells and showed induction of CHOP and stabilized ATF2 transcription factor, whose expression was high in FBXO32 mutant hearts. They conclude that ER stress, abnormal CHOP activation and CHOP-induced apoptosis with no UPR effector activation underlie the FBXO32 mutation induced cardiomyopathy.

The rare p.Gly243Arg variant in FXBO32 was identified previously by Al-Hassnan et al. in a consanguineous family, where 4 homozygous siblings were affected with cardiomyopathy, while 6 heterozygote carriers (4 siblings and their parents) and a negative sibling were unaffected, supporting the conclusion that this mutation is recessive (1). Although authors did not mention the initial report, the reviewer assumed that the authors obtained explant heart samples from the IV.5 subject (homozygous carrier) and his sibling IV.7 (heterozygous carrier) and compared them to heart samples of patients with idiopathic dilated cardiomyopathy and control individuals. Although the idea of testing the transcriptomics in those patients is interesting and the authors put a significant effort and time in elucidating the molecular mechanisms of familial cardiomyopathy in this family, the study has left many alarming questions that the authors should resolve.

1. What is nature of the Gly243Arg-FXBO32 variant? Is it homozygous recessive or dominant negative? The authors should address this question first and clarify why they are pursuing this study. Unfortunately, the reviewer could not find the critical information for the patients tested (homozygous

vs heterozygous) in this paper.

2. In order to claim that the Gly243Arg-FBXO32 is a causal mutation, authors should test the heterozygote state by transfecting the WT and mutant FBXO32 mixture in the cells. Alternatively, the authors should mention that heterozygote inheritance pattern as a limitation of their studies at least.

3. Authors are suggested to perform a rigorous genetic testing of two patients studied in this study at least by whole exome sequencing in order to clearly define the causality of the FBXO32 variant. As a reviewer I believe, the multigenic inheritance may underlie the disease in this family, while the FBXO32 variant may act as a genetic modifier that increases the penetrance and severity of the disease in this family.

4. As outcome of #1-3 points, a medical and genetic relevance of the paper is uncertain despite impressive molecular biology experimental support for the FBXO32 function. The authors are required to perform additional experiments to improve the clinical relevance and quality of the manuscript. Ideally if possible, all members (affected and unaffected) including heterozygous carriers and negative individuals are indicated for exome sequencing. All major and minor comments that supported this conclusion are detailed below.

Major comments:

1. As mentioned above, reviewer is concerned about the causal nature of this variant for early onset heart failure. Authors need to do a whole exome sequencing of the IV.7 heterozygous patient as there are many obstacles with IV.7 subject.

- Authors should reference the initial report by Al-Hassnan et al or clearly inform the family genetic background.

- Authors need to separate the 2 patients' studied data. Analysis and discussion should be held separately as they are not genetically same for the FBXO32 inheritance.

2. Despite the IV.7 subject is a heterozygote carrier per Al-Hassnan et al, he shows significantly higher levels of ATF4 (Fig.2a) and active Caspase 3 (Fig.3a) and lower BCL2 compared to the IV.5 subject. This data does not support (may be challenges) the main idea of the paper that "mutation in FBXO32 causes dilated cardiomyopathy through upregulation of ER-stress mediated apoptosis".

3. In addition to cardiomyopathy, authors mention that FBXO32 is a muscle specific and cause muscle atrophy and premature aging. However, no skeletal muscle or premature aging phenotype is observed in this family. In my opinion, authors need to focus on finding a cardiac-specific cause to provide a strong argument why the family has only cardiac phenotype.

4. Authors started the Results section with the findings on impaired mitochondrial function in FBXO32 hearts with no introductory mitochondrial connections to the main idea of the paper. This is somewhat is confusing because mitochondrial abnormalities are common and general features in many types of cardiomyopathies.

5. In addition to confusing writing style, there are many obstacles that the readers will be confused further. For example: in the heart of one patient from an unrelated family carrying another genetic mutation causing DCM???.

For example: Line 144: "the heart samples clustered into four groups with IDC hearts displaying similar gene expression pattern to that from the heart of Family B". If the Family B is included in the equation for any statement like that, the authors should at least mention what mutation or what gene is affected in order to draw this conclusion and why Family A is distinct from others.

6. "Consistent with the role of FBXO32 in heart failure, toxicity annotation in IPA showed that cardiac hypertrophy ranked number one in the mutant.." Authors are asked to clarify an association of FBXO32 with all these cardiac phenotypes, including hypertrophy.

7. Authors note that mitochondrial function and apoptotic cell death are dominant processes altered in the heart of the patients carrying the FBXO32 mutation. Then, suddenly, authors change the direction

and the ER stress becomes a critical pathway. Authors should justify why they preferred an ER stress pathway for their studies, despite the data in Fig.1 shows no ER stress in the top pathways and signaling.

Saying that, the statement: "Pathways found to be altered in the mutant FBXO32 heart from our transcriptional analysis included the ER or ER stress response, suggesting a critical role of the ER stress pathway in the pathogenesis of dilated cardiomyopathy in patients carrying the FBXO32 mutation" is arguable because ER stress can be secondary to genetic assault not of FBXO32 variant, especially in IV.7 patient (heterozygote carrier).

8. Again, a confusing statement (line 182): "For this analysis, we were able to include the heart from another patient of Family A with the FBXO32 mutation..." Authors should provide a clear reference about any subject participated in the study and his (her) genetic background (heterozygous or homozygous).

9. Line 193: "These results show activation of ER-stress apoptosis in patient hearts with the FBXO32 mutation." There are no any supporting data provided to this statement. First, the authors should do TUNEL staining in these patient hearts to visualize and confirm an apoptosis. Next, the authors should show there are no other apoptotic intrinsic and extrinsic pathways are activated (at least TNF α , FAS, and cyt C as mitochondria are concerned).

10. Line 211: "...suggest that the ER-stress pathway is a major contributor to the cardiomyopathy caused by the FBXO32 mutation". Heart failure in dilated cardiomyopathy is a result of contractile dysfunction of cardiac muscle working as a constant pump. Therefore, authors should include analysis of contractile and sarcomeric proteins from transcriptome data they have and perform experiments on a protein level.

11. Line 220: "Western blot analysis showed increased level of cleaved caspase-3 in the two hearts carrying the FBXO32 mutation." As I see, there is more significant difference in active caspase-3 between the 2 hearts carrying the FBXO32 mutation. Please explain this disparity. Caspase-3 cleavage is not specific to CHOP apoptosis, it is activated downstream to many apoptotic pathways mentioned above. Thus again, authors should exclude all other pathways that caused caspase-3 cleavage.

12. Line: 258: "ATF2 was efficiently precipitated with WT- and mutant FBXO32. Significant interaction of ATF2 with FBXO32 was also observed in human hearts..." Extensive IP studies were performed, however, authors should support these results with immunohistochemical analysis to visualize localization and overlap of the proteins in human heart samples to functionally confirm.

Minor comments:

1. The last part of the Introduction (Lines 114-12) describes the results of the study and is better to move to the Discussion section.

2. "Skp-cullin-F-Box protein (SCF) complex" (Line 108) and "SKP1/CUL1/ROC1 complex" (line 131) are same? Please organize all terminology and their abbreviations throughout the manuscript text.

3. Lines 176-180 is better to omit as all these are somewhat repeating the introduction.

4. All Westerns should have a statistical analysis.

5. Lines 303-304: please omit. This should be described in the methods section.

6. There repeats between introduction and discussion sections. Please eliminate those repeat and condense both sections.

REFERENCE:

1. Al-Hassnan ZN, Shinwari ZM, Wakil SM et al. A substitution mutation in cardiac ubiquitin ligase, FBXO32, is associated with an autosomal recessive form of dilated cardiomyopathy. *BMC Med Genet* 2016;17:3.

Dear reviewers,

We are pleased to submit our revised manuscript entitled “*Mutation in FBXO32 Causes Dilated Cardiomyopathy Through Upregulation of ER-Stress Mediated Apoptosis*”. We would like to thank the reviewers for their interest in our work and insightful comments. We performed additional experiments to address the major issues identified and have revised our manuscript accordingly. Changes are highlighted in blue color in the revised manuscript. A point-by-point response to each of the reviewers’ comment is provided below.

Reviewer 1 (Remarks to the Author):

The manuscript by Al-Yacoub et al investigates the consequences of FBXO32 mutation on dilated cardiomyopathy. The authors perform RNA-seq on a single patient sample, and determine that ER-stress-related genes are altered when compared to controls, or other causes of dilated cardiomyopathy. Subsequent investigations on the same sample and on a second sample from the same family confirm, by PCR and Western blot, that markers of ER stress are up-regulated in these patients, which results in activated caspase 3 and apoptosis. The data are interesting and seems to point to a role for this mutation to induce an ER stress response that is mediated by interaction with ATF2. The biggest issue with the study is the reliance on one patient sample, even if confirmation is done in samples from two individuals. At least the cell experiments should have been repeated minimally 4 times, to determine if the observed effects are truly due to the mutation.

Response: We thank the reviewer for acknowledging the relevance of our paper. The major issue pointed out by the reviewer was the low number of replicates for the cell experiments. We agree it is important to establish new mechanisms in a rigorous way. As asked by the reviewer, we increased the number of biological replicates in all cell experiments. The **new Figure 4e** shows the quantitative analysis of ATF2 protein expression in HEK293 cells expressing WT- or mutant-FBXO32 from 4 independent experiments. The data shows significantly higher ATF2 protein expression in HEK293 cells expressing mutant-FBXO32 compared to cells expressing WT-FBXO32.

We performed additional experiments to support the observation that FBXO32 regulates ATF2 protein expression. Using co-immunoprecipitation, we show that ATF2 ubiquitination is reduced by the mutant-FBXO32. This new data, from 3 independent transfections for both FBXO32-WT and FBXO32-mutant (n=2-3 replicates), is shown in the **new Figure 5c** and is described in the revised manuscript in **line 259-268**.

We increased the sample number for the Western blots showing CHOP-mediated apoptosis. Quantitative analysis from 5 independent experiments shows a significant increase of CHOP and BIM_{EL} expression in mutant-FBXO32 cells compared to WT-FBXO32 cells (**New Figure 6b&c**). In addition, we performed another apoptosis assay to show that the *FBXO32* mutation enhances apoptosis. Consistent with our Western blot analysis, ELISA assay also shows a significantly higher apoptosis in cells expressing mutant-FBXO32 compared to WT-FBXO32 expressing cells. This new data is included in the **new Figure 6a** and is described in **line 274-276** of the revised manuscript.

1 – In the text the authors mention that GADD34 is increased in the hearts of the mutant patients, but bands in Figure 2 are not well defined. An upper band is only present in the mutant hearts, but a lower band is dramatically decreased. There is no explanation for the lower band or how it relates to the mutation. Could the upper band be p-GADD34? Otherwise, give MW marker and say that the lower band is non-specific?

Response: The reviewer diligently noticed 2 bands for GADD34 Western blot in the human hearts. The upper low migrating band is consistent with the expected 100 KDa molecular weight of GADD34 as indicated by the company the antibody was purchased from (Proteintech). As suggested by the reviewer, the lower more prominent migrating band is non-specific. We have now added an asterisk in the margin of the **New Figure 2a** to clearly indicate this non-specific band and have modified the figure legend accordingly.

2 – The significance of the isoforms of BIM should be discussed.

Response: Our Western blot analysis shows that Bim_{EL} is robustly increased in mutant *FBXO32* hearts, while the other BIM isoforms (Bim_L and Bim_S) are either not detected or expressed at relatively low-level in all the human hearts. BIM has been reported to inhibit autophagy independent of its proapoptotic function by interacting with Beclin-1 in a complex with LC8. Among the three Bim isoforms, only Bim_{EL} and Bim_L strongly interact with Beclin-1 (51). In addition, Bim_{EL} is an important factor regulating the switch from autophagy to apoptosis during prolonged starvation (52). Based on these reports, we speculate that the upregulation of Bim_{EL} in *FBXO32* mutant hearts could provide a link between the activation of apoptosis and the impairment of autophagy that we

previously documented in the mutant *FBXO32* hearts (37). In the revised manuscript, we clarify that only BimEL is increased in *FBXO32* mutant hearts (**line 204**) and as suggested by the reviewer, we comment on the significance of the BimEL isoform in the discussion section (**line 354-362**).

3 – The small sample size needs to be added to a limitations section.

Response: We agree with the reviewer that the 2 hearts with the *FBXO32* mutation is a limitation to the study. As requested by the reviewer, we have added a Limitation Section (**line 614-621**) as follows: “*One limitation of our study is with the availability of only two human hearts with the FBXO32 mutation. Because sample heterogeneity is often an issue with human specimens, we minimized the collecting time of the human hearts by freezing the human hearts in liquid nitrogen immediately after surgical removal from the patients, to preserve tissue integrity. Using this strict protocol, variability between explanted hearts was reduced to a minimum. In addition, the effect of the mutation was studied in HEK293 cells, which validated the results observed in the mutant human hearts.*”

4 – Is there a reason why protein levels in Figure 2 were not compared to Family B?

Response: We thank the reviewer for this question. We included Family B only in Figure 1 to document that the transcriptional profiling between 2 different families with different genetic mutations causing DCM are distinct. Our goal was *not* to characterize the cardiomyopathy of Family B in this paper. It is being pursued in a separate study. The distinct gene expression profiling between Family A and Family B gives us additional confidence that the induction of ER-stress apoptosis is specific to the dysfunctional SCF^{FBXO32}-E3 ligase caused by the *FBXO32* mutation and not a general response of DCM.

5 – Please name the excel supplementary files as described in the text.

Response: As requested by the Reviewer, the labelling of the Supplementary files now match the descriptions in the manuscript.

6 – Bar graphs are not appropriate, especially since this is a N=2. Dot blots should be shown.

Response: We thank the Reviewer for this valid point. We now represent the two mutant *FBXO32* hearts as two separate samples in the quantitative analysis (**New Figure 2b, New Figure 3b and New Figure 4c**).

7 – I assume that the two samples from family A in Figure 1 are technical replicates from the same heart. This needs to be disclosed in the figure legends.

Response: As requested by the reviewer, we clearly indicate in the **Revised Figure 1 legend** that the samples from Family A and Family B are from technical replicates (**line 875-876**).

8 – The small sample size decreases rigor of the work. Although this is a rare mutation, the authors do provide cell data on the effect of the mutation in cells. It is absolutely necessary that the authors increase the number of cell experiments so that rigor can be increased through proper statistical analysis. As is, this is a subtle suggestion of the role of this mutation on ER stress and apoptosis.

Response: As indicated before, **the new Figure 4d&e** showing ATF2 protein expression in HEK293 cells expressing WT- or mutant-*FBXO32*, is representative of 5 independent experiments. The **new Figure 5c** showing that *FBXO32* regulates ATF2 ubiquitination is from 3 independent transfections including duplicate or triplicate samples. Results showing that the *FBXO32* mutation enhances apoptosis measured by ELISA assay and CHOP-mediated apoptosis measured by Western blot are from 7 independent experiments (**New Figure 6**).

Minor revisions

Line 90-change sever—severe

Line 840-change heats—hearts

In Figure 4C change label from protein expression to ATF2 expression

Response: These minor changes are now included in the Revised manuscript and in the **New Figure 4c&e**.

Reviewer #2 (Remarks to the Author):

In the manuscript entitled “Mutation in FBXO32 causes dilated cardiomyopathy 1 through upregulation of ER-stress mediated Apoptosis”, Al-Yacoub et al found a new mutant of Fbxo32 (Gly243Arg) and this single amino acid mutant led to aberrant activation of CHOP and the CHOP associated apoptosis. They also found that ATF2 expression level was elevated in the Fbxo32 mutant heart. They also confirmed the results in 293 cells by overexpressing the mutant Fbxo32. They found an interesting Fbxo32 mutant and its potential functions in heart. I have the following concerns for the authors to be addressed.

1. There are many key experiments performed in 293 cells. The advantage of using 293 cells has been stated by the authors clearly that no endogenous Fbxo32 to compete with the mutant Fbxo32. However, there may be potential problems by solely doing experiments in 293 cells. The context of 293 cells is drastically different from that of cardiac cells. The key results such as the interaction between Fbxo32 and ATF2, the induction of ATF2 after overexpression of mutant Fbxo32 should be confirmed in cardiac cells.

Response: We thank the Reviewer for recognizing the significance of our paper and for his/her suggestions. We agree with the importance to confirm the key results from HEK293 cells in cardiac cells. Because primary cardiac cells are difficult to transfect with plasmid DNA, we generated recombinant lentiviruses for WT-FBXO32 and mutant-FBXO32 using the 3rd generation lentiviral vectors. Prior to performing experiments in primary neonatal rat ventricular myocytes (NRVM), we validated that WT- and mutant FBXO32 were transduced at similar high-levels in 293T cells (see **panel a** of the **adjacent Figure**). Unfortunately, after multiple attempts, experiments failed in NRVM. The mutant-FBXO32 protein was consistently expressed at a much lower level than the WT-FBXO32 (see **panel b** of the **adjacent Figure**). This precluded us from assessing the effect of the mutant protein on ATF2 and ER-stress apoptosis. We attribute this result to the high rate of protein degradation that occurs in cardiac cells and particularly when mutant proteins are expressed. In spite of this, we were able to show co-localization of endogenous ATF2 with WT-FBXO32 in NRVM using confocal microscopy, which is included as a new result in **Supplementary Fig. 5**. Altogether, the following results point towards ATF2 and the ER-stress as major drivers of the cardiomyopathy due to the *FBXO32* mutation:

- We provide evidence that ATF2 is strongly up-regulated in the two mutant hearts as well as in HEK293 cells expressing the mutant-FBXO32 protein. This coincides with the up-regulation of the ER-stress apoptosis marker CHOP, increased apoptosis measured by ELISA assay and increased BimEL protein expression measured by Western blot analysis.
- A strong interaction between ATF2 and FBXO32 is detected in the human heart. Using confocal microscopy, we show that ATF2 co-localizes with FBXO32 in cardiac cells.
- Consistent with the function of FBXO32 as an E3 ubiquitin ligase, we provide evidence that FBXO32 regulates ATF2 ubiquitination. Co-immunoprecipitation experiments show reduced ATF2 ubiquitination in cells expressing mutant-FBXO32.

2. The ubiquitination status of ATF2 after Fbxo32 mutant or WT Fbxo32 overexpression should be checked by ubiquitination Western blot.

Response: We thank the Reviewer for suggesting this experiment. As requested, we expressed WT-FBXO32 or mutant-FBXO32 in HEK293 cells together with ATF2 and HA-ubiquitin. After immunoprecipitation of ATF2 with a specific antibody, Western blot analysis using an anti-ubiquitin antibody shows significant ubiquitination of ATF2 in the presence of WT-FBXO32, which is reduced in the presence of mutant-FBXO32. This result suggests

that the *FBXO32* mutation impairs the SCF complex and as a consequence, ubiquitination of ATF2 is reduced which in turn reduces its degradation. This new result is included in the **new Figure 5c** and in the revised manuscript **line 259-268**.

3. To determine the occurrence of apoptosis only by Western blot of cleaved caspase 3 is kind of weak. Other assays such as Annexin-V staining, TUNEL assays should be performed to confirm the occurrence of apoptosis in *Fbxo32* mutant cardiac cells.

Response: We agree with the Reviewer that additional apoptosis assays were important to strengthen our conclusion. Because we did not have access to human heart sections, we could not measure apoptosis using Annexin-V or TUNEL staining in the human hearts. Instead, we used an ELISA assay (Cell Death Detection ELISA kit, Roche Diagnostics) and provide new data showing a significantly higher apoptosis in HEK293 cells expressing mutant *FBXO32* compared to WT-*FBXO32*. This new data is included in the **new Figure 6a** and in **line 274-276** of the revised manuscript.

4. Similarly, Other assays such as Annexin-V staining, TUNEL assays should be performed to confirm the occurrence of apoptosis in 293 cells overexpressing mutant or WT *Fbxo32*.

Response: As indicated above, we measured apoptosis in HEK293 cells expressing WT- or mutant *FBXO32* using an ELISA-based apoptosis assay. Results show significantly higher apoptosis in cells expressing mutant-*FBXO32* compared to cells expressing WT-*FBXO32*. This new data is included in the **new Figure 6a** and in the revised manuscript (**line 274-276**).

5. The authors described in the text that “untransfected cells that were used as negative control, exhibited the same expression level for CHOP and BIM as cells overexpressing WT-*FBXO32* upon DTT treatment.” The corresponding figure has not been cited.

Response: We added the corresponding figure (**new Figure 6b&c**) in **line 286** of the revised manuscript.

6. The expression level of GADD34 and Caspase 3 seem to be unstable in untransfected 293 cells. What is the explanation? Is it simply experimental variations? Why is the variation specifically big with these two proteins?

Response: we agree that GADD34 protein expression varies from one transfection experiment to another. We attribute this variability to the cell density which can differ between experiments. HEK293 cells have a high replication rate and in spite of our biggest effort, it is difficult to seed the cells at the same density prior to transfection. An alternative explanation for this variability is the passage number of the cells which is different between experiments.

7. Related to question 6, in Fig. 5a expt 1, GADD34 protein level decreased when compared to the untransfected 293 cells; while GADD34 protein level increased when compared to the untransfected 293 cells in expt2. Please explain.

Response: please see Response 6.

Reviewer #3 (Remarks to the Author):

The manuscript entitled: "Mutation in FBXO32 Causes Dilated Cardiomyopathy Through Upregulation of ER-Stress Mediated Apoptosis" by Al-Yacoub and colleagues, describes their original research focused on investigation the mechanism by which the FBXO32 mutation (identified previously) causes advanced cardiomyopathy. They performed transcriptional profiling and biochemical assays in the heart of patients with the FBXO32 mutation and compared with heart samples obtained from patients with idiopathic dilated cardiomyopathy and control donors. They found reduced activation of the three UPR effectors while a strong up-regulation of the transcription factor CHOP and of its target genes. They also claim that they found severe apoptosis specifically in FBXO32 mutant hearts. Then, they expressed mutant FBXO32 in cells and showed induction of CHOP and stabilized ATF2 transcription factor, whose expression was high in FBXO32 mutant hearts. They conclude that ER stress, abnormal CHOP activation and CHOP-induced apoptosis with no UPR effector activation underlie the FBXO32 mutation induced cardiomyopathy.

The rare p.Gly243Arg variant in FXBO32 was identified previously by Al-Hassnan et al. in a consanguineous family, where 4 homozygous siblings were affected with cardiomyopathy, while 6 heterozygote carriers (4 siblings and their parents) and a negative sibling were unaffected, supporting the conclusion that this mutation is recessive (1). Although authors did not mention the initial report, the reviewer assumed that the authors obtained explant heart samples from the IV.5 subject (homozygous carrier) and his sibling IV.7 (heterozygous carrier) and compared them to heart samples of patients with idiopathic dilated cardiomyopathy and control individuals. Although the idea of testing the transcriptomics in those patients is interesting and the authors put a significant effort and time in elucidating the molecular mechanisms of familial cardiomyopathy in this family, the study has left many alarming questions that the authors should resolve.

Response: We thank the Reviewer for recognizing the extensive functional work that we performed to provide a new mechanism by which the *FBXO32-G243R* mutation causes dilated cardiomyopathy. The Reviewer refers to a publication from the Hassnan group in which patient IV.5 and IV.7 are respectively homozygous and heterozygous for the mutation. We would like to point out that in our study, patient IV.5 and IV.7, are both homozygous for the *FBXO32* mutation. We refer the Reviewer to our original manuscript (Al-Yacoub et al., Genome Biology, 2016) in which we clearly describe the nature of the mutation in each of the subjects. The difference noticed by the Reviewer between our study and the Hassnan study is simply due to the different labelling of the subjects. We realize that we may not have been clear enough about the homozygosity of the *FBXO32* mutation of the two patient hearts used in the current study. We now have added this information in the Summary, Introduction, Results and Discussion sections of the revised manuscript. We also share the family pedigree with the Reviewer (see **pedigree below**), which clearly shows that patients IV.5 and IV.7 are homozygous for the *FBXO32* mutation.

1. What is nature of the Gly243Arg-FXBO32 variant? Is it homozygous recessive or dominant negative? The authors should address this question first and clarify why they are pursuing this study. Unfortunately, the reviewer could not find the critical information for the patients tested (homozygous vs heterozygous) in this paper.

Response: We thank the Reviewer for pointing out this deficiency. As mentioned before, we have now clarified that the *FBXO32*-G243R mutation is homozygous recessive (lines 39, 107, 127-128, 302) and refer the readers to our original paper (Al-Yacoub et al., 2016) which includes the clinical features of the patients as well as the genetic and functional characterization of the mutation.

2. In order to claim that the Gly243Arg-FXBO32 is a causal mutation, authors should test the heterozygote state by transfecting the WT and mutant *FBXO32* mixture in the cells. Alternatively, the authors should mention that heterozygote inheritance pattern as a limitation of their studies at least.

Response: Subjects with the heterozygous *FBXO32* mutation (e.g. the parents and unaffected siblings) do *not* develop cardiomyopathy. We agree with the reviewer that testing the heterozygous state would be an excellent experiment to establish causality of a heterozygous mutation. However, this approach cannot be used to test the effect of the *FBXO32* mutation because it is homozygous.

3. Authors are suggested to perform a rigorous genetic testing of two patients studied in this study at least by whole exome sequencing in order to clearly define the causality of the *FBXO32* variant. As a reviewer I believe, the multigenic inheritance may underlie the disease in this family, while the *FBXO32* variant may act as a genetic modifier that increases the penetrance and severity of the disease in this family.

Response: We agree with the Reviewer that DCM usually has different degree of penetrance and expressivity especially in case of autosomal dominant modes of transmission. In the case of the family that we recruited, it is a consanguineous family from Saudi Arabia for which we already established that the cardiomyopathy is monogenic (Al-Yacoub et al., 2016). Indeed, after recruiting this large family including the parents and the 10 siblings, we determined that the mode of inheritance was autosomal recessive. Our rigorous genetic analysis revealed a single region of homozygosity only shared between the affected siblings and which did not include any known cardiomyopathy gene. In addition, linkage analysis revealed a single peak with a maximum LOD score which matched the region of homozygosity. After performing next generation sequencing on one affected patient and filtration of the variants, only one homozygous variant survived (*FBXO32* NM_058229.3:c., 727G > C, p.Gly243Arg) and was found to be pathogenic using 3 different algorithms. We also know that the affected patients developed the cardiomyopathy at a very young age. In fact, all the affected patients were referred for heart transplant before the age of 28. Altogether, as shown in our original work (Al-Yacoub et al., 2016), these results show that for this family, the cardiomyopathy is a monogenic disorder caused by the *FBXO32* mutation.

4. As outcome of #1-3 points, a medical and genetic relevance of the paper is uncertain despite impressive molecular biology experimental support for the *FBXO32* function. The authors are required to perform additional experiments to improve the clinical relevance and quality of the manuscript. Ideally if possible, all members (affected and unaffected) including heterozygous carriers and negative individuals are indicated for exome sequencing. All major and minor comments that supported this conclusion are detailed below.

Response: We are thankful to the reviewer for acknowledging the extent of our molecular work. As mentioned before, the affected patients were all homozygous for the *FBXO32* mutation. The clinical evaluations of the patients were done by experienced cardiologists at the Heart Centre at King Faisal Specialist Hospital & Research Centre in Riyadh, the largest referring hospital in the entire middle East region. Most importantly, our original work already went through a rigorous peer-reviewed process which established its genetic and clinical relevance.

Major comments:

1. As mentioned above, reviewer is concerned about the causal nature of this variant for early onset heart failure. Authors need to do a whole exome sequencing of the IV.7 heterozygous patient as there are many obstacles with IV.7 subject.

Response: As mentioned before, patient IV.7 is homozygous for the mutation, not heterozygous.

- Authors should reference the initial report by Al-Hassnan et al or clearly inform the family genetic background.

Response: We realize that we may have not been clear enough about the nature of the *FBXO32* mutation for the 2 patients included in the present study. To remedy this, we clearly indicate that both patients are homozygous carriers in the Revised manuscript (**lines 39, 107, 127-128, 302**). For the clinical features of the patients, we refer the reader to our original discovery (Al-Yacoub et al., 2016), **line 301-303 and 428-430**.

- Authors need to separate the 2 patients' studied data. Analysis and discussion should be held separately as they are not genetically same for the *FBXO32* inheritance.

Response: As mentioned before, the two patients included in our study are both homozygous carriers.

2. Despite the IV.7 subject is a heterozygote carrier per Al-Hassnan et al, he shows significantly higher levels of ATF4 (Fig.2a) and active Caspase 3 (Fig.3a) and lower BCL2 compared to the IV.5 subject. This data does not support (may be challenges) the main idea of the paper that "mutation in *FBXO32* causes dilated cardiomyopathy through upregulation of ER-stress mediated apoptosis".

Response: As indicated before, patient IV.7 is homozygous for the *FBXO32* mutation. As the other homozygous carrier (patient IV.5), patient IV.7 shows increased BimEL and cleaved PARP1 compared to idiopathic dilated hearts whereas BCL-2 is reduced. Active caspase-3 is more robustly increased in patient IV.7, which is likely due to a biological difference. Overall, however, both patients IV.5 and IV.7 have an increase apoptosis response.

3. In addition to cardiomyopathy, authors mention that *FBXO32* is a muscle specific and cause muscle atrophy and premature aging. However, no skeletal muscle or premature aging phenotype is observed in this family. In my opinion, authors need to focus on finding a cardiac-specific cause to provide a strong argument why the family has only cardiac phenotype.

Response: We thank the Reviewer for this comment, which is an excellent point that we also contemplated. The *FBXO32* protein is enriched in skeletal and cardiac muscle. Studies performed in muscle tissue from rodents initially supported a role of *FBXO32* in skeletal muscle atrophy mainly due to the increase of *FBXO32* in models of muscle atrophy and the identification of the myogenic factor MyoD as a *FBXO32* substrate. However, deletion of *FBXO32* in mice attenuates atrophy but does not prevent it and forced expression of *FBXO32* does not induce muscle atrophy, indicating that other proteins are regulating skeletal muscle mass. In contrast, loss and gain-of-function studies in mice revealed a role of *FBXO32* in physiological and pathological cardiac hypertrophy and cardiomyopathy (Usui S et al., *Circ Res* 2011; Adams V et al., *Cardiovascular Res* 2007; Zaglia T et al., *JCI* 2014). Our published study showed that the *FBXO32*-G243R mutation does *not* alter *FBXO32* protein expression but instead, abrogates binding to its partner proteins of the SCF complex (Al-Yacoub, 2016). Based on this and on the clinical features of the affected patients with the *FBXO32* mutation (heart failure with no reported skeletal muscle abnormalities), the cardiac defects are consistent with the major role of *FBXO32* in the heart and that cardiomyopathy is the major disease caused by the *FBXO32* mutation.

4. Authors started the Results section with the findings on impaired mitochondrial function in *FBXO32* hearts with no introductory mitochondrial connections to the main idea of the paper. This is somewhat is confusing because mitochondrial abnormalities are common and general features in many types of cardiomyopathies.

Response: We thank the Reviewer for this comment. We changed the heading of this section to "Major pathways dysregulated in mutant *FBXO32* hearts" because pathways other than Mitochondrial dysfunction are altered in mutant *FBXO32* hearts (e.g. cardiac hypertrophy, protein ubiquitination, etc...) as shown in Figure 1.

5. In addition to confusing writing style, there are many obstacles that the readers will be confused further. For example: in the heart of one patient from an unrelated family carrying another genetic mutation causing DCM???

For example: Line 144: "the heart samples clustered into four groups with IDC hearts displaying similar gene expression pattern to that from the heart of Family B". If the Family B is included in the equation for any statement like that, the authors should at least mention what mutation or what gene is affected in order to draw this conclusion and why Family A is distinct from others.

Response: As mentioned before, we included Family B to simply document that the transcriptional profiling between two different families with different genetic mutations causing DCM are very distinct. Our goal was *not* to characterize the cardiomyopathy of Family B in this paper. It is being pursued in a separate study.

6. “Consistent with the role of FBXO32 in heart failure, toxicity annotation in IPA showed that cardiac hypertrophy ranked number one in the mutant..” Authors are asked to clarify an association of FBXO32 with all these cardiac phenotypes, including hypertrophy.

Response: We performed IPA from control, mutant *FBXO32* and idiopathic dilated hearts to identify, in an *unbiased way*, pathways significantly dysregulated and unique to the mutant *FBXO32* hearts. Subsequently and because of the novelty, we intentionally focused on the ER-stress pathway and performed deep molecular work to establish its role in FBXO32-mediated cardiomyopathy. As noticed by the Reviewer, there are additional known and yet to be characterized pathways implicated in the cardiomyopathy caused by the *FBXO32* mutation. Because it is not feasible to pursue all dysregulated pathways, we focused on a selected one for detailed mechanistic work, which is customary in studies involving transcriptional profiling.

7. Authors note that mitochondrial function and apoptotic cell death are dominant processes altered in the heart of the patients carrying the FBXO32 mutation. Then, suddenly, authors change the direction and the ER stress becomes a critical pathway. Authors should justify why they preferred an ER stress pathway for their studies, despite the data in Fig.1 shows no ER stress in the top pathways and signaling.

Response: As indicated in the previous point, we performed transcriptional profiling to identify new pathways implicated in the *FBXO32* cardiomyopathy. As expected, many pathways were significantly dysregulated in the mutant *FBXO32* heart among which was the ER-stress pathway (**Supplementary Data 1 and 2**). Excited by this result and because it is simply impossible to focus on all dysregulated pathways, we purposefully focused on characterizing the ER-stress pathway, which turned out to be critical in the *FBXO32*-mediated cardiomyopathy.

Saying that, the statement: “Pathways found to be altered in the mutant FBXO32 heart from our transcriptional analysis included the ER or ER stress response, suggesting a critical role of the ER stress pathway in the pathogenesis of dilated cardiomyopathy in patients carrying the FBXO32 mutation” is arguable because ER stress can be secondary to genetic assault not of FBXO32 variant, especially in IV.7 patient (heterozygote carrier).

Response: We thank the Reviewer for raising this important point. It is true that ER-stress is activated in many cardiac diseases. However, if it was a secondary effect of the cardiomyopathy, the ER-stress pathway would also have been dysregulated in the IDC hearts, which is not the case. In addition, expression of the mutant-FBXO32 in HEK293 cells was sufficient to induce ER-stress apoptosis. Altogether, these results support the view that the ER-stress response is *specific* to the patient with the *FBXO32* mutation and a cause rather than a consequence of the cardiomyopathy.

8. Again, a confusing statement (line 182): “For this analysis, we were able to include the heart from another patient of Family A with the FBXO32 mutation...” Authors should provide a clear reference about any subject participated in the study and his (her) genetic background (heterozygous or homozygous).

Response: as indicated previously, details on the family and on the *FBXO32* mutation can be found in our original Genome Biology paper and in our revised manuscript.

9. Line 193: “These results show activation of ER-stress apoptosis in patient hearts with the FBXO32 mutation.” There are no any supporting data provided to this statement. First, the authors should do TUNEL staining in these patient hearts to visualize and confirm an apoptosis. Next, the authors should show there are no other apoptotic intrinsic and extrinsic pathways are activated (at least TNF α , FAS, and cyt C as mitochondria are concerned).

Response: We thank the Reviewer for the suggestion to confirm the apoptosis result using more than one assay. We followed his/her recommendations and measured apoptosis in cells expressing WT- and mutant FBXO32 using an ELISA-based assay. Results show enhanced apoptosis in cells expressing mutant-FBXO32 and are shown in the **new Figure 6a**. Regarding the assessment of additional apoptotic pathways, the goal of our study

was not to investigate apoptosis in general, but to focus on a specific type of cell death called ER-stress-associated apoptosis. This specific type of cell death occurs through the activation of the mitochondrial-apoptosis pathway via regulation of BH3-Only Protein. Therefore, we intentionally focused on proteins implicated in the ER-stress and *not* on the role of the extrinsic apoptosis pathway.

10. Line 211: "...suggest that the ER-stress pathway is a major contributor to the cardiomyopathy caused by the FBXO32 mutation". Heart failure in dilated cardiomyopathy is a result of contractile dysfunction of cardiac muscle working as a constant pump. Therefore, authors should include analysis of contractile and sarcomeric proteins from transcriptome data they have and perform experiments on a protein level.

Response: As mentioned before, it is not realistic to perform functional work on all dysregulated pathways. The primary focus of our study was the ER-stress pathway.

11. Line 220: "Western blot analysis showed increased level of cleaved caspase-3 in the two hearts carrying the FBXO32 mutation." As I see, there is more significant difference in active caspase-3 between the 2 hearts carrying the FBXO32 mutation. Please explain this disparity. Caspase-3 cleavage is not specific to CHOP apoptosis, it is activated downstream to many apoptotic pathways mentioned above. Thus again, authors should exclude all other pathways that caused caspase-3 cleavage.

Response: As noticed by the Reviewer, the Western blot in Figure 3 indeed shows a stronger up-regulation of active caspase-3 in patient IV.7 compared to patient IV.5, which indicates biological differences between the two patients. We agree that the availability of 2 human hearts in our study is a limitation. We included a "Limitation of the study" section in **line 612-619** of the revised manuscript.

12. Line: 258: "ATF2 was efficiently precipitated with WT- and mutant FBXO32. Significant interaction of ATF2 with FBXO32 was also observed in human hearts..." Extensive IP studies were performed, however, authors should support these results with immunohistochemical analysis to visualize localization and overlap of the proteins in human heart samples to functionally confirm.

Response: We thank the Reviewer for the suggestion to validate the results of co-immunoprecipitation using a different approach. We performed immunofluorescence confocal microscopy in primary neonatal rat cardiomyocytes (NRVM). Because NRVM are difficult to transfect with plasmid DNA, we generated a recombinant lentivirus expressing WT-FBXO32 using the 3rd generation lentiviral vectors. Confocal microscopy shows a co-localization of WT-FBXO32 with endogenous ATF2. This new result is included in **Supplementary Fig. 5**.

Minor comments:

1. The last part of the Introduction (Lines 114-12) describes the results of the study and is better to move to the Discussion section.

Response: It is quite common to include a brief description of the major findings at the end of the introduction to give the readers the major findings of the study and ease the reading of the manuscript. We agree that the summary was too long in the first version and have shortened it in the Revised version of the manuscript (**line 111-120**).

2. "Skp-cullin-F-Box protein (SCF) complex" (Line 108) and "SKP1/CUL1/ROC1 complex" (line 131) are same? Please organize all terminology and their abbreviations throughout the manuscript text.

Response: We thank the Reviewer for noticing this discrepancy. We now provide the same terminology throughout the Revised manuscript.

3. Lines 176-180 is better to omit as all these are somewhat repeating the introduction.

Response: As requested by the Reviewer, we removed these sentences as they were already in the introduction.

4. All Westerns should have a statistical analysis.

Response: We included graphs for all the Western blots showing averages and standard deviations when appropriate.

5. Lines 303-304: please omit. This should be described in the methods section.

Response: As requested, we removed this sentence.

6. There repeats between introduction and discussion sections. Please eliminate those repeat and condense both sections.

Responses: We removed redundant sentences between the introduction and the discussion in the Revised manuscript.

REFERENCE:

1. Al-Hassnan ZN, Shinwari ZM, Wakil SM et al. A substitution mutation in cardiac ubiquitin ligase, FBXO32, is associated with an autosomal recessive form of dilated cardiomyopathy. BMC Med Genet 2016;17:3.

Reviewers' comments:

Reviewer #1 (Remarks to the Author):

In general the authors have significantly improved the manuscript.

Two points only:

All graphs should show individual experiments instead of a bar graph showing just the mean and the SD.

I am still confused about the tissue samples. In the response to reviewers the authors say that the samples come from two different hearts, but also mention they are technical replicates. Are these two samples from two different individuals or are they from the same heart that was loaded twice? That needs to be clarified.

Reviewer #2 (Remarks to the Author):

The authors answered all my questions and I have no further concerns.

Reviewer #3 (Remarks to the Author):

This is a second review of the manuscript entitled: "Mutation in FBXO32 Causes Dilated Cardiomyopathy Through Upregulation of ER-Stress Mediated Apoptosis" by Al-Yacoub and colleagues, describes their original research focused on investigation the mechanism by which the FBXO32 mutation (identified previously) causes advanced cardiomyopathy. The paper is improved in due to clarifying the mutation state and additional biological experiments.

Comment #1. The authors shared the pedigree of the family A in their response, however the reviewer didn't find it in the revised manuscript. The authors advised to display the pedigree in the manuscript so the reader with no access to the previous publications can see which patients are the referred as IV.5 and IV.7 indexes.

Comment #2. In order to claim that the Gly243Arg-FXBO32 is a causal mutation, authors should test the heterozygote state by transfecting the WT and mutant FBXO32 mixture in the cells. Alternatively, the authors should mention that heterozygote inheritance pattern as a limitation of their studies at least.

Review: The authors didn't respond to this comment, and this is noted to be important by other reviewer also. The authors should perform the cellular studies with heterozygote cells and the negative results will validate the pathogenicity of homozygous mutation.

Comment #3. Authors are suggested to perform a rigorous genetic testing of two patients studied in this study at least by whole exome sequencing in order to clearly define the causality of the FBXO32 variant. As a reviewer I believe, the multigenic inheritance may underlie the disease in this family, while the FBXO32 variant may act as a genetic modifier that increases the penetrance and severity of the disease in this family.

Review: The authors performed a whole exome sequencing on one patient, which is great. However, no results of genetic analysis on all variants and SNPs identified is not provided in the revised manuscript. As a standard, at least six (6) in silico predictions should be provide to claim that the variant is pathogenic.

Comment #4. As outcome of #1-3 points, a medical and genetic relevance of the paper is uncertain despite impressive molecular biology experimental support for the FBXO32 function. The authors are required to perform additional experiments to improve the clinical relevance and quality of the manuscript. Ideally if possible, all members (affected and unaffected) including heterozygous carriers and negative individuals are indicated for exome sequencing. All major and minor comments that supported this conclusion are detailed below.

Review: Authors should clearly state in the manuscript as: The clinical evaluations of the patients were done by experienced cardiologists and their genetic and clinical relevance has been reported (REFERENCE)

Please insert this statement in the background section to keep the readers in the loop of the paper from the beginning.

Major comments:

1. Responded appropriately. Great.

2. Despite the IV.7 subject is a heterozygote carrier per Al-Hassnan et al, he shows significantly higher levels of ATF4 (Fig.2a) and active Caspase 3 (Fig.3a) and lower BCL2 compared to the IV.5 subject. This data does not support (may be challenges) the main idea of the paper that "mutation in FBXO32 causes dilated cardiomyopathy through upregulation of ER-stress mediated apoptosis".

Author's Response: As indicated before, patient IV.7 is homozygous for the FBXO32 mutation. As the other homozygous carrier (patient IV.5), patient IV.7 shows increased BimEL and cleaved PARP1 compared to idiopathic dilated hearts whereas BCL-2 is reduced. Active caspase-3 is more robustly increased in patient IV.7, which is likely due to a biological difference. Overall, however, both patients IV.5 and IV.7 have an increase apoptosis response.

Review: As the biological differences are mentioned, these observations need to be discussed in the Discussion section. That is why the authors should display the results of whole exome sequencing and comment the results on the observed biological (genetic) differences between siblings who carry the identical homozygous FBXO32 mutation.

3. In addition to cardiomyopathy, authors mention that FBXO32 is a muscle specific and cause muscle atrophy and premature aging. However, no skeletal muscle or premature aging phenotype is observed in this family. In my opinion, authors need to focus on finding a cardiac-specific cause to provide a strong argument why the family has only cardiac phenotype.

Responded appropriately. Great.

4. Authors started the Results section with the findings on impaired mitochondrial function in FBXO32 hearts with no introductory mitochondrial connections to the main idea of the paper. This is somewhat is confusing because mitochondrial abnormalities are common and general features in many types of cardiomyopathies.

Review: Line 305: Please remove "impaired mitochondrial function". The authors found altered mitochondrial genes expression, but they did not assess the actual function in this work.

5. In addition to confusing writing style, there are many obstacles that the readers will be confused further. For example: in the heart of one patient from an unrelated family carrying another genetic mutation causing DCM???.

For example: Line 144: "the heart samples clustered into four groups with IDC hearts displaying similar gene expression pattern to that from the heart of Family B". If the Family B is included in the

equation for any statement like that, the authors should at least mention what mutation or what gene is affected in order to draw this conclusion and why Family A is distinct from others.

Author's Response: As mentioned before, we included Family B to simply document that the transcriptional profiling between two different families with different genetic mutations causing DCM are very distinct. Our goal was not to characterize the cardiomyopathy of Family B in this paper. It is being pursued in a separate study.

Review: The editor notes this is a biased inclusion of the Family B. The family B is advised to be removed or the statement that family B does not carry the FBXO32 mutation. Please explain why Familial B in the figure 1a has 4 columns if only 1 patient is included from the Family B. What is a difference between Familial B and Family B? Please provide the N number for all samples in all figures as needed.

6. "Consistent with the role of FBXO32 in heart failure, toxicity annotation in IPA showed that cardiac hypertrophy ranked number one in the mutant.." Authors are asked to clarify an association of FBXO32 with all these cardiac phenotypes, including hypertrophy.

Response: We performed IPA from control, mutant FBXO32 and idiopathic dilated hearts to identify, in an unbiased way, pathways significantly dysregulated and unique to the mutant FBXO32 hearts. Subsequently and because of the novelty, we intentionally focused on the ER-stress pathway and performed deep molecular work to establish its role in FBXO32-mediated cardiomyopathy. As noticed by the Reviewer, there are additional known and yet to be characterized pathways implicated in the cardiomyopathy caused by the FBXO32 mutation. Because it is not feasible to pursue all dysregulated pathways, we focused on a selected one for detailed mechanistic work, which is customary in studies involving transcriptional profiling.

Review: Reviewer asked to explain why hypertrophy (as the top in IPA) and other pathways are not selected. You can discuss the results based on your transcriptomic results in the discussion section and express why hypertrophy (or other pathways listed in figure 1d) has not been chosen vs ER stress, or may be both (or all pathways identified) related with each other. The brief discussion will make the paper more convincing why the particular ER – apoptosis pathway has been selected for molecular studies.

7. Authors note that mitochondrial function and apoptotic cell death are dominant processes altered in the heart of the patients carrying the FBXO32 mutation. Then, suddenly, authors change the direction and the ER stress becomes a critical pathway. Authors should justify why they preferred an ER stress pathway for their studies, despite the data in Fig.1 shows no ER stress in the top pathways and signaling.

Saying that, the statement: "Pathways found to be altered in the mutant FBXO32 heart from our transcriptional analysis included the ER or ER stress response, suggesting a critical role of the ER stress pathway in the pathogenesis of dilated cardiomyopathy in patients carrying the FBXO32 mutation" is arguable because ER stress can be secondary to genetic assault not of FBXO32 variant, especially in IV.7 patient (heterozygote carrier).

Review: Please see above.

8. Again, a confusing statement (line 182): "For this analysis, we were able to include the heart from another patient of Family A with the FBXO32 mutation.." Authors should provide a clear reference about any subject participated in the study and his (her) genetic background (heterozygous or homozygous).

Review: Please see above.

9. Line 193: "These results show activation of ER-stress apoptosis in patient hearts with the FBXO32 mutation." There are no any supporting data provided to this statement. First, the authors should do TUNEL staining in these patient hearts to visualize and confirm an apoptosis. Next, the authors should show there are no other apoptotic intrinsic and extrinsic pathways are activated (at least TNF α , FAS, and cyt C as mitochondria are concerned).

Response: We thank the Reviewer for the suggestion to confirm the apoptosis result using more than one assay. We followed his/her recommendations and measured apoptosis in cells expressing WT- and mutant FBXO32 using an ELISA-based assay. Results show enhanced apoptosis in cells expressing mutant-FBXO32 and are shown in the new Figure 6a. Regarding the assessment of additional apoptotic pathways, the goal of our study was not to investigate apoptosis in general, but to focus on a specific type of cell death called ER-stress-associated apoptosis. This specific type of cell death occurs through the activation of the mitochondrial-apoptosis pathway via regulation of BH3-Only Protein. Therefore, we intentionally focused on proteins implicated in the ER-stress and not on the role of the extrinsic apoptosis pathway.

Review Response: The authors partially responded to this comment. TUNEL is a crucial to visualize the apoptotic cells and an apoptotic end-product of DNA fragmentation, which authors did not perform. To discuss the activation of other extrinsic and intrinsic pathways, the transcriptome data should be sufficient to explain (simply see the levels of TNF α , FAS, and cyt C or other genes involved vs ER-stress genes).

10. Line 211: "...suggest that the ER-stress pathway is a major contributor to the cardiomyopathy caused by the FBXO32 mutation". Heart failure in dilated cardiomyopathy is a result of contractile dysfunction of cardiac muscle working as a constant pump. Therefore, authors should include analysis of contractile and sarcomeric proteins from transcriptome data they have and perform experiments on a protein level.

Response: As mentioned before, it is not realistic to perform functional work on all dysregulated pathways. The primary focus of our study was the ER-stress pathway.

Review Response: Absolutely agree with authors that functional work on all dysregulated pathways is not feasible. However, the authors have already obtained transcriptomic data and the differentially expressed genes are clear on patients. Please include a brief discussion on sarcomeric genes as the reviewer believes that these genes are disturbed and this data would be interesting for readers as well.

11. Line 220: "Western blot analysis showed increased level of cleaved caspase-3 in the two hearts carrying the FBXO32 mutation." As I see, there is more significant difference in active caspase-3 between the 2 hearts carrying the FBXO32 mutation. Please explain this disparity. Caspase-3 cleavage is not specific to CHOP apoptosis, it is activated downstream to many apoptotic pathways mentioned above. Thus again, authors should exclude all other pathways that caused caspase-3 cleavage.

Please see above.

12. Line: 258: "ATF2 was efficiently precipitated with WT- and mutant FBXO32. Significant interaction of ATF2 with FBXO32 was also observed in human hearts..." Extensive IP studies were performed, however, authors should support these results with immunohistochemical analysis to visualize localization and overlap of the proteins in human heart samples to functionally confirm.

Responded properly, great.

Point-by-point response to Reviewer 1 and Reviewer 3. Changes made to the previous version are highlighted in blue color in the revised manuscript.

Reviewer 1

In general the authors have significantly improved the manuscript.

Two points only:

All graphs should show individual experiments instead of a bar graph showing just the mean and the SD.

Response: We agree that representing individual values for the hearts of the 3 groups (Control, FDC and IDC) would provide consistency. Nevertheless, to comply with the space limitation of the figures, we would like to keep the current representation. This is a personal preference which does not impact on the results.

I am still confused about the tissue samples. In the response to reviewers the authors say that the samples come from two different hearts, but also mention they are technical replicates. Are these two samples from two different individuals or are they from the same heart that was loaded twice? That needs to be clarified.

Response: We apologize for the confusion regarding the technical and biological replicates of the human hearts. We have now clarified that for Family A and B, samples are from one heart run in duplicates and quadruplicates respectively. Control and idiopathic hearts are from 3 and 4 different individuals respectively. This information has been added to the **new Figure 1 legend (lines 904-906)**

Reviewer 3

This is a second review of the manuscript entitled: "Mutation in FBXO32 Causes Dilated Cardiomyopathy Through Upregulation of ER-Stress Mediated Apoptosis" by Al-Yacoub and colleagues, describes their original research focused on investigation the mechanism by which the FBXO32 mutation (identified previously) causes advanced cardiomyopathy. The paper is improved in due to clarifying the mutation state and additional biological experiments.

Comment #1. The authors shared the pedigree of the family A in their response, however the reviewer didn't find it in the revised manuscript. The authors advised to display the pedigree in the manuscript so the reader with no access to the previous publications can see which patients are the referred as IV.5 and IV.7 indexes.

Response: As asked by the Reviewer, we included the family pedigree in the Supplementary Figures (see **New Supplementary Fig. 1**). Old supplementary Fig. 1,2,3,4,5 and 6 are now re-labelled as Supplementary Fig. 2,3,4,5,6 and 7 respectively in the revised manuscript.

Comment #2. In order to claim that the Gly243Arg-FXBO32 is a causal mutation, authors should test the heterozygote state by transfecting the WT and mutant FBXO32 mixture in the cells. Alternatively, the authors should mention that heterozygote inheritance pattern as a limitation of their studies at least.

Review: The authors didn't respond to this comment, and this is noted to be important by other reviewer also. The authors should perform the cellular studies with heterozygote cells and the negative results will validate the pathogenicity of homozygous mutation.

Response: we thank the Reviewer for suggesting an alternative to testing the heterozygote nature of the mutation. We now indicate in the section "Limitation to the study" that "*the heterozygous state of the mutation was not tested in the current study because our genetic analysis showed that only the patients with the homozygous mutation develop DCM. It would be of interest to assess the effect of the heterozygous mutation in cells. This is beyond the scope of this work and will be the topic of future investigations*" (lines 643-647).

Comment #3. Authors are suggested to perform a rigorous genetic testing of two patients studied in this study at least by whole exome sequencing in order to clearly define the causality of the FBXO32 variant. As a reviewer I believe, the multigenic inheritance may underlie the disease in this family, while the FBXO32 variant may act as a genetic modifier that increases the penetrance and severity of the disease in this family.

Review: The authors performed a whole exome sequencing on one patient, which is great. However, no results of genetic analysis on all variants and SNPs identified is not provided in the revised manuscript. As a standard, at least six (6) in silico predictions should be provide to claim that the variant is pathogenic.

Response: As requested by the Reviewer we performed additional *in silico* predictions to strengthen our genetic analysis documenting the pathogenicity of the *FBXO32* mutation. Our original manuscript included 4 *in silico* analysis (PolyPhen-2, MutationTaster, SIFT and PHRED) that all predicted that the variant is pathogenic or disease causing (Al-Yacoub, 2016). We performed 3 additional *in silico* predictions using CADD, GERP and REVEL, which gave scores of 25.8, 4.95 and 0.658 respectively. These all confirmed that the *FBXO32* variant is disease causing or deleterious. This information has been added to the revised manuscript (lines 134-138).

Comment #4. As outcome of #1-3 points, a medical and genetic relevance of the paper is uncertain despite impressive molecular biology experimental support for the FBXO32 function. The authors are required to perform additional experiments to improve the clinical relevance and quality of the manuscript. Ideally if possible, all members (affected and unaffected) including heterozygous carriers and negative individuals are indicated for exome sequencing. All major and minor comments that supported this conclusion are detailed below.

Review: Authors should clearly state in the manuscript as: The clinical evaluations of the patients were done by experienced cardiologists and their genetic and clinical relevance has been reported (REFERENCE)

Please insert this statement in the background section to keep the readers in the loop of the paper from the beginning.

Response: As requested by the Reviewer, we included a sentence “*The clinical evaluations of the patients were done by experienced cardiologists and their genetic and clinical relevance has been reported*” and the reference **lines 129-130**.

Major comments:

1. Responded appropriately. Great.

2. Despite the IV.7 subject is a heterozygote carrier per Al-Hassnan et al, he shows significantly higher levels of ATF4 (Fig.2a) and active Caspase 3 (Fig.3a) and lower BCL2 compared to the IV.5 subject. This data does not support (may be challenges) the main idea of the paper that “mutation in FBXO32 causes dilated cardiomyopathy through upregulation of ER-stress mediated apoptosis”.

Author’s Response: As indicated before, patient IV.7 is homozygous for the FBXO32 mutation. As the other homozygous carrier (patient IV.5), patient IV.7 shows increased BimEL and cleaved PARP1 compared to idiopathic dilated hearts whereas BCL-2 is reduced. Active caspase-3 is more robustly increased in patient IV.7, which is likely due to a biological difference. Overall, however, both patients IV.5 and IV.7 have an increase apoptosis response.

Review: As the biological differences are mentioned, these observations need to be discussed in the Discussion section. That is why the authors should display the results of whole exome sequencing and comment the results on the observed biological (genetic) differences between siblings who carry the identical homozygous FBXO32 mutation.

Response: As requested by the Reviewer, we added a sentence “*Active caspase-3 was more robustly increased in patient IV.7, likely due to a biological difference*” (**lines 382-384**) of the revised manuscript.

3. In addition to cardiomyopathy, authors mention that FBXO32 is a muscle specific and cause muscle atrophy and premature aging. However, no skeletal muscle or premature aging phenotype is observed in this family. In my opinion, authors need to focus on finding a cardiac-specific cause to provide a strong argument why the family has only cardiac phenotype.

Responded appropriately. Great.

4. Authors started the Results section with the findings on impaired mitochondrial function in FBXO32 hearts with no introductory mitochondrial connections to the main idea of the paper. This is somewhat is confusing because mitochondrial abnormalities are common and general features in many types of cardiomyopathies.

Review: Line 305: Please remove “impaired mitochondrial function”. The authors found altered mitochondrial genes expression, but they did not assess the actual function in this work.

Response: We modified the sentence as “*associated with altered expression of mitochondrial genes*” **line 318**.

5. In addition to confusing writing style, there are many obstacles that the readers will be confused further. For example: in the heart of one patient from an unrelated family carrying another genetic mutation causing DCM???.

For example: Line 144: “the heart samples clustered into four groups with IDC hearts displaying similar gene expression pattern to that from the heart of Family B”. If the Family B is included in the equation for any statement like that, the authors should at least mention what mutation or what gene is affected in order to draw this conclusion and why Family A is distinct from others.

Author’s Response: As mentioned before, we included Family B to simply document that the transcriptional profiling between two different families with different genetic mutations causing DCM are very distinct. Our goal was not to characterize the cardiomyopathy of Family B in this paper. It is being pursued in a separate study.

Review: The editor notes this is a biased inclusion of the Family B. The family B is advised to be removed or the statement that family B does not carry the FBXO32 mutation. Please explain why Familial B in the figure 1a has 4 columns if only 1 patient is included from the Family B. What is a difference between Familial B and Family B? Please provide the N number for all samples in all figures as needed.

Response: We apologize for the confusion and have now clarified that for Family A and B, samples are from one heart run in duplicates and quadruplicates respectively. Control and idiopathic hearts are from 3 and 4 different individuals respectively. This information has been added to the **new Fig. 1 legend (lines 904-906)**.

6. “Consistent with the role of FBXO32 in heart failure, toxicity annotation in IPA showed that cardiac hypertrophy ranked number one in the mutant..” Authors are asked to clarify an association of FBXO32 with all these cardiac phenotypes, including hypertrophy.

Response: We performed IPA from control, mutant FBXO32 and idiopathic dilated hearts to identify, in an unbiased way, pathways significantly dysregulated and unique to the mutant FBXO32 hearts. Subsequently and because of the novelty, we intentionally focused on the ER-stress pathway and performed deep molecular work to establish its role in FBXO32-mediated cardiomyopathy. As noticed by the Reviewer, there are additional known and yet to be characterized pathways implicated in the cardiomyopathy caused by the FBXO32 mutation. Because it is not feasible to pursue all dysregulated pathways, we focused on a selected one for detailed mechanistic work, which is customary in studies involving transcriptional profiling.

Review: Reviewer asked to explain why hypertrophy (as the top in IPA) and other pathways are not selected. You can discuss the results based on your transcriptomic results in the discussion section and express why hypertrophy (or other pathways listed in figure 1d) has not been chosen vs ER stress, or may be both (or all pathways identified) related with each other. The brief discussion will make the paper more convincing why the particular ER – apoptosis pathway has been selected for molecular studies.

Response: The activation of hypertrophy genes is a well-known response in DCM including familial DCM, so we did not focus on hypertrophy genes dysregulated in the mutant *FBXO32* hearts. Rather, we found it more interesting to focus on less well characterized pathways like the ER stress, which recently has emerged as a prominent pathway playing a role in heart failure. Indeed, the ER is a critical organelle which communicates with mitochondria to ensure the maintenance of cellular homeostasis. Because pathways found to be altered in the mutant *FBXO32* heart from our transcriptional analysis included the ER or ER stress response, we performed deep molecular work to address its role in the cardiomyopathy due to the *FBXO32* mutation.

7. Authors note that mitochondrial function and apoptotic cell death are dominant processes altered in the heart of the patients carrying the FBXO32 mutation. Then, suddenly, authors change the direction and the ER stress becomes a critical pathway. Authors should justify why they preferred an ER stress pathway for their studies, despite the data in Fig.1 shows no ER stress in the top pathways and signaling. Saying that, the statement: “Pathways found to be altered in the mutant FBXO32 heart from our transcriptional analysis included the ER or ER stress response, suggesting a critical role of the ER stress pathway in the pathogenesis of dilated cardiomyopathy in patients carrying the FBXO32 mutation” is

arguable because ER stress can be secondary to genetic assault not of FBXO32 variant, especially in IV.7 patient (heterozygote carrier).

Review: Please see above.

Response: Same as Response to point 6.

8. Again, a confusing statement (line 182): “For this analysis, we were able to include the heart from another patient of Family A with the FBXO32 mutation...” Authors should provide a clear reference about any subject participated in the study and his (her) genetic background (heterozygous or homozygous).

Review: Please see above.

Response: As asked by the Reviewer, we clarify that the second heart sample was from patient IV.5, who was homozygous for the *FBXO32* mutation, **line 181-182** of the revised manuscript.

9. Line 193: “These results show activation of ER-stress apoptosis in patient hearts with the FBXO32 mutation.” There are no any supporting data provided to this statement. First, the authors should do TUNEL staining in these patient hearts to visualize and confirm an apoptosis. Next, the authors should show there are no other apoptotic intrinsic and extrinsic pathways are activated (at least TNF α , FAS, and cyt C as mitochondria are concerned).

Response: We thank the Reviewer for the suggestion to confirm the apoptosis result using more than one assay. We followed his/her recommendations and measured apoptosis in cells expressing WT- and mutant FBXO32 using an ELISA-based assay. Results show enhanced apoptosis in cells expressing mutant-FBXO32 and are shown in the new Figure 6a. Regarding the assessment of additional apoptotic pathways, the goal of our study was not to investigate apoptosis in general, but to focus on a specific type of cell death called ER-stress-associated apoptosis. This specific type of cell death occurs through the activation of the mitochondrial-apoptosis pathway via regulation of BH3-Only Protein. Therefore, we intentionally focused on proteins implicated in the ER-stress and not on the role of the extrinsic apoptosis pathway.

Review Response: The authors partially responded to this comment. TUNEL is a crucial to visualize the apoptotic cells and an apoptotic end-product of DNA fragmentation, which authors did not perform. To discuss the activation of other extrinsic and intrinsic pathways, the transcriptome data should be sufficient to explain (simply see the levels of TNF α , FAS, and cyt C or other genes involved vs ER-stress genes).

Response: We searched for genes of the extrinsic apoptosis pathway in our transcriptome data. We could not find a significant dysregulation of TNF α , TNFR1, DR3/4/5, FAS, FADD, TRADD, RIP or DED, the most important members of this pathway. Only TNFRSF21 (DR6) was differentially regulated in the mutant *FBXO32* heart. Although we cannot exclude the possibility that differences may occur at the protein level, this suggests that the extrinsic apoptosis pathway plays a minor role in *FBXO32*-mediated cardiomyopathy. We added this information to the discussion section of the revised manuscript (**lines 388-394**).

10. Line 211: “...suggest that the ER-stress pathway is a major contributor to the cardiomyopathy caused by the FBXO32 mutation”. Heart failure in dilated cardiomyopathy is a result of contractile dysfunction of cardiac muscle working as a constant pump. Therefore, authors should include analysis of contractile and sarcomeric proteins from transcriptome data they have and perform experiments on a protein level.

Response: As mentioned before, it is not realistic to perform functional work on all dysregulated pathways. The primary focus of our study was the ER-stress pathway.

Review Response: Absolutely agree with authors that functional work on all dysregulated pathways is not feasible. However, the authors have already obtained transcriptomic data and the differentially expressed

genes are clear on patients. Please include a brief discussion on sarcomeric genes as the reviewer believes that these genes are disturbed and this data would be interesting for readers as well.

Response: The survey of genes encoding sarcomeric proteins in our microarray data showed that TTN, MYBPC3, TNNT3, MYL2, MYL6B, TMOD1 and TMOD3 are significantly dysregulated in the mutant *FBXO32* heart. These proteins play important roles in cardiac contraction. Therefore, there dysregulation may contribute to the cardiomyopathy due to the *FBXO32* mutation. It would be interesting to assess whether *FBXO32*, as a ubiquitin ligase, regulate their expression. We tried to add this paragraph to the revised manuscript. However, we found it was a distraction to the focus of the paper, the ER-stress. For this reason, we did not include it to the revised manuscript.

11. Line 220: “Western blot analysis showed increased level of cleaved caspase-3 in the two hearts carrying the *FBXO32* mutation.” As I see, there is more significant difference in active caspase-3 between the 2 hearts carrying the *FBXO32* mutation. Please explain this disparity. Caspase-3 cleavage is not specific to CHOP apoptosis, it is activated downstream to many apoptotic pathways mentioned above. Thus again, authors should exclude all other pathways that caused caspase-3 cleavage.

Response: Please see above Point 9 response.

12. Line: 258: “ATF2 was efficiently precipitated with WT- and mutant *FBXO32*. Significant interaction of ATF2 with *FBXO32* was also observed in human hearts...” Extensive IP studies were performed, however, authors should support these results with immunohistochemical analysis to visualize localization and overlap of the proteins in human heart samples to functionally confirm.

Responded properly, great.